# A cryo-EM structure of KTF1-bound polymerase V transcription elongation complex

Hong-Wei Zhang[1,2,4], Kun Huang [1,2,4], Zhan-Xi Gu[1,2,4], Xiao-Xian Wu[1], Jia-Wei Wang[3] & Yu Zhang [1]✉

De novo DNA methylation in plants relies on transcription of RNA polymerase V (Pol V) along with KTF1, which produce long non-coding RNAs for recruitment and assembly of the DNA methylation machinery. Here, we report a cryo-EM structure of the Pol V transcription elongation complex bound to KTF1. The structure reveals the conformation of the structural motifs in the active site of Pol V that accounts for its inferior RNA-extension ability. The structure also reveals structural features of Pol V that prevent it from interacting with the transcription factors of Pol II and Pol IV. The KOW5 domain of KTF1 binds near the RNA exit channel of Pol V providing a scaffold for the proposed recruitment of Argonaute proteins to initiate the assembly of the DNA methylation machinery. The structure provides insight into the Pol V transcription elongation process and the role of KTF1 during Pol V transcription-coupled DNA methylation.

DNA methylation at the 5′ position of cytosine is essential for gene regulation, transposon silencing and imprinting in both animals and plants[1–4]. As such, growing lines of evidence has shown that DNA methylation plays critical roles in plant development and physiology, such as immunity[5], genome stability[6], tracheary element differentiation[7], and sexual development[8,9]. Plant DNA methylation occurs in all cytosine sequence contexts: CG, CHG and CHH (H represents A, T or C)[10].

In plants, de novo DNA methylation is mediated through the RNA-directed DNA methylation (RdDM) pathway[2]. The RdDM pathway can be divided into two major steps: production of siRNA and targeted DNA methylation. So far, two routes have been identified for the siRNA production: the canonical RNA polymerase IV (Pol IV)/ RNA-dependent RNA polymerase 2 (RDR2)-mediated route and the non-canonical Pol II/RDR6-mediated route[11–14]. In the canonical Pol IV/ RDR2 route, Pol IV and RDR2 form a stable two-polymerase complex that produces double-stranded RNA (dsRNA) through a "back-tracking-triggered RNA channeling" mechanism[15–18]. The dsRNA is subsequently processed by DICER-LIKE PROTEIN3 (DCL3) and loaded into ARGONAUTE 4/6 (AGO4/6)[19–23]. The noncanonical Pol II/RDR6-

RdDM route is initiated by Pol II transcription at transposon elements (TE) in heterochromatic and intergenic regions[24]. The single-stranded TE mRNA is subsequently converted by RDR6 to dsRNA[25], which is further processed by DCL2/3/4 and AGO6[26].

The targeted DNA methylation is the second major step of the RdDM pathway. Pol V transcription elongation complex functions as the assembly scaffold for targeted DNA methylation, and therefore plays a central role in RdDM pathway. The siRNA-loaded AGO4/6 is recruited to Pol V transcription elongation complex (TEC) through base-paring with Pol V-derived scaffold RNA and interacting with the repeated motifs containing glycine-tryptophan/tryptophan-glycine (GW/WG) motifs of the Pol V-NRPE1 subunit and/or of the AGO4/6-recruiting factor KOW domain-containing transcription factor 1 (KTF1; also named SPT5L)[27–32]. The DNA methyltransferase DOMAINS REARRANGED METHYLASE2 (DRM2) is subsequently recruited to the Pol V TEC and add methyl modification on the Pol V transcribed genomic DNA[33,34].

Pol V is evolutionarily derived from Pol II through gene duplication[35]. In *Arabidopsis thaliana*, Pol V and Pol II share 7 subunits including NRP(B/D/E)3, NRP(A/B/C/D/E)6, NRP(A/B/C/D/E)8,

[1]Key Laboratory of Synthetic Biology, CAS Center for Excellence in Molecular Plant Sciences, Shanghai Institute of Plant Physiology and Ecology, Chinese Academy of Sciences, Shanghai 200032, China. [2]University of Chinese Academy of Sciences, Beijing 100049, China. [3]National Key Laboratory of Plant Molecular Genetics, CAS Center for Excellence in Molecular Plant Sciences, Shanghai Institute of Plant Physiology and Ecology, Chinese Academy of Sciences, Shanghai 200032, China. [4]These authors contributed equally: Hong-Wei Zhang, Kun Huang, and Zhan-Xi Gu. ✉e-mail: yzhang@cemps.ac.cn

NRP(B/D/E)9, NRP(A/B/C/D/E)10, NRP(B/D/E)11, and NRP(A/B/C/D/E) 12; Pol V and Pol IV share 2 subunits, including NRP(D/E)2 and NRP(D/E) 4; and Pol V has 3 unique subunits, including NRPE1, NRPE5, NRPE7 (Supplementary Table 1). The unique subunit composition of Pol V provides basis for its special catalytic activity, regulatory factors, and cellular function. Compared with Pol II, Pol V exhibits lower RNA elongation rate and higher fidelity[36], the slowed elongation rate might provide windows for recruitment of AGO4/6 while the higher fidelity ensures the subsequent precise base-pairing of AGO4/6 loaded siRNA and Pol V transcript[36]. The two unique largest subunits of Pol V, which constitute the active-site cleft, likely account for its unique catalytic activities.

The transcribed genomic regions and associated general transcription factors of Pol V are also substantially different from those of Pol II. Pol V typically produces scaffold RNA at regions with transposon elements and DNA repeats[29,37], while Pol II mainly produces mRNA at gene-coding regions. A recent report suggests Pol V performs pervasive transcription at much broader genomic regions to surveil the genome[38]. The unique transcription properties of Pol V are likely also attributed to its specific general transcription factors that guide Pol V to the distinct genomic loci to initiate transcription and to couple DNA methylation during its elongation[11–14]. The potential general transcription initiation factors of Pol V include the DDR complex, comprising DEFECTIVE IN MERISTEM SILENCING 3 (DMS3), DEFECTIVE IN RNA-DIRECTED DNA METHYLATION (DRD1), and DEFECTIVE IN RNA-DIRECTED DNA METHYLATION 1 (RDM1), and the SUVH2/9 complex, the latter of which binds methylated DNA and likely recruits the DDR complex and Pol V[27,32,39–41]. However, it is unknown how DDR facilitates Pol V to unwind genomic DNA and initiate RNA synthesis. The transcription factors of Pol V at the transcription elongation stage include KTF1 and SPT4[28,42,43]. None of Pol V-specific termination factors have been discovered and it is still unknown how Pol V terminates its RNA synthesis.

KTF1 is a paralog of SPT5, the only universally conserved transcription elongation factor in all three domains of lives (NusG in bacteria and SPT5 in archaea and eukaryotes). Depletion of its encoding gene *RDM3* in Arabidopsis reduced DNA methylation levels at the RdDM loci but did not affect the levels of 24-nt siRNA, suggesting KTF1 functions at the downstream stage of the RdDM pathway[28,42]. Arabidopsis KTF1 contains an NGN domain, three KOW domains (KOW1/4/5), and a long C-terminal tail comprising 42 GW/WG repeats[28]. It was reported that the C-terminal GW/WG-repeat domain of KTF1 and the C-terminal GW/WG-repeat domain of NRPE1 play redundant role in recruiting AGO4/6[44], as retaining only one of the domains shows normal DNA methylation profile. However, the other domains of KTF1 may play roles beyond the AGO4/6 recruitment as depletion of the full-length KTF1 protein in *rdm3* exhibits substantial defects in Pol V transcript slicing and DNA methylation[28,29]. The cryo-EM structure of SPT4/5-bound Pol II TEC reveals that the conserved NGN domain forms a heterodimer with SPT4 and binds at the upstream dsDNA channel of Pol II[45,46]. The conserved KOW1, KOW4, and KOW 5 domains of SPT5 bind near the distal upstream dsDNA channel, stalk, and RNA exit channel, respectively. SPT5 has been reported to facilitate Pol II elongation by stabilizing the elongation complex and recruiting other Pol II elongation factors[47]. It is intriguing that Pol V transcript levels were barely affected in the *rdm3* mutant[28], raising the possibility that KTF1 might not affect the elongation of Pol V but simply functions as a AGO4/6-recruting factor.

To sum up, RdDM is a complex epigenetic pathway requiring two plant-specific Pol II-derived DNA-dependent RNA polymerases (*i.e.* Pol IV and Pol V) as well as a number of specialized factors. However, due to the lack of structural information of Pol V, it remains unclear how Pol V assembles with its unique subunits, interacts with its transcription factors, and works together with them to deposit targeted DNA methylation. Here, we present the cryo-EM structure of Arabidopsis KTF1-bound transcription elongation complex of Pol V (KTF1-bound Pol V TEC). The structure shows that Pol V accommodates elongation RNA-DNA scaffold in a similar manner as Pol II, Pol V possess unique structural features that prevents it from interacting general transcription factors of other polymerases, and Pol V interacts the KTF1-KOW5 domain near its RNA exit channel.

## Results

### Structure determination of the KTF1-bound Pol V TEC

The epitope-tagged Pol V was purified from *Arabidopsis thaliana* T87 cells stably expressing 3xFLAG-tagged NRPE1, the largest subunit of Pol V (Supplementary Fig. 1). LC-MS/MS analysis of the purified complex shows the presence of nine Pol V subunits (Supplementary Data 1 and Supplementary Table 2), in line with the previous identification of Pol V subunits from Arabidopsis callus except that NRPE5c is also identified in our epitope-tagged Pol V besides NRPE5a[48]. Our current map could not distinguish the two subunits and NRPE5a is modeled into the structure. The discrepancy is likely due to different experimental materials used for Pol V preparation. The additional NRP(D/E)4, NRP(A/B/C/D/E)6 and NRP(A/B/C/D/E)12 subunits were absent in the LC-MS/MS but present in the cryo-EM map (see below). The LC-MS/MS analysis result shows peptide signals for the three Pol V-specific subunits NRPE1, NRPE5, and NRPE7, consistent with the previous report[48]. The epitope-tagged Pol V is catalytically active in extending an RNA primer (Supplementary Fig. 1d). The *Arabidopsis thaliana* KTF1 (residues 1–712, the conserved N-terminal domain) and SPT4 were co-expressed in *E. coli* cells and the two proteins form a stable heterodimer (Supplementary Fig. 1e, f).

To assemble KTF1-SPT4-bound Pol V TEC, we reconstituted the complex using *Arabidopsis thaliana* Pol V core enzyme, the KTF1-SPT4 heterodimer, and a nucleic-acid scaffold comprising a pre-melted transcription bubble and a 30-nt RNA with 9-nt region complementary to the template DNA (Fig. 1a, b). We subsequently applied the KTF1-SPT4-bound Pol V TEC complex for cryo-EM data collection and determined the structures using a single-particle cryo-EM method (Supplementary Fig. 2 and Supplementary Table 3). Multiple-cycle 3D classification and refinement resulted in a major class and a minor class of single particles, from which two cryo-EM maps at resolutions 3.2 Å and 4.3 Å were constructed, respectively. The lower-resolution map shows more complete signal for the stalk--NRP(D/E)4 and NRPE7 subunits--and clamp domains (Supplementary Fig. 2). Focused refinement around the KTF1-KOW5 domain and NRPE5 subunit resulted in two local maps at 3.0 Å and 3.7 Å, respectively, allowing for confident modeling of the active site of Pol V, the KTF1-KOW5 domain, and the NRPE1-NRPE5 interface (Supplementary Fig. 2).

### The protein-DNA/RNA interactions in the KTF1-bound Pol V TEC

The cryo-EM maps show strong signals for all 12 subunits of Pol V (Fig. 1d), 19-bp downstream DNA in the downstream dsDNA binding channel, 9-bp RNA-DNA hybrid in the active-site cleft, and 6-bp upstream dsDNA in the upstream dsDNA channel (Fig. 1a, c). Although the KTF1-SPT4 heterodimer was included in the complex reconstitution, the cryo-EM map only shows signal for the KOW5 domain of KTF1 bound near the RNA exit channel (Fig. 1f). Pol V folds into the typical crab-claw shape of multiple-subunit DNA-dependent RNA polymerases (Fig. 1e). Structural superimposition between *Arabidopsis thaliana* Pol V TEC and yeast Pol II TEC reveals significant similarity of the overall structure of the two polymerases (Fig. 1g). Although a similar nucleic-acid scaffold was used for structure determination, the main DNA cleft of Pol V TEC is substantially widened compared with that of Pol II TEC, due to outward rotation of the lobe domain of NRP(D/E)2 and the clamp domain of NRPE1 subunit (Fig. 1g, h).

The Pol V-TEC structure reveals detailed interaction between Pol V and the RNA-DNA hybrid in the active-site cleft (Fig. 2a, b). The RNA-DNA hybrid of the transcription bubble in the active site cleft adopts the post-translocation state with the 'i + 1' site empty for binding of the incoming NTP (Figs. 1c, 2a, b). The phosphate

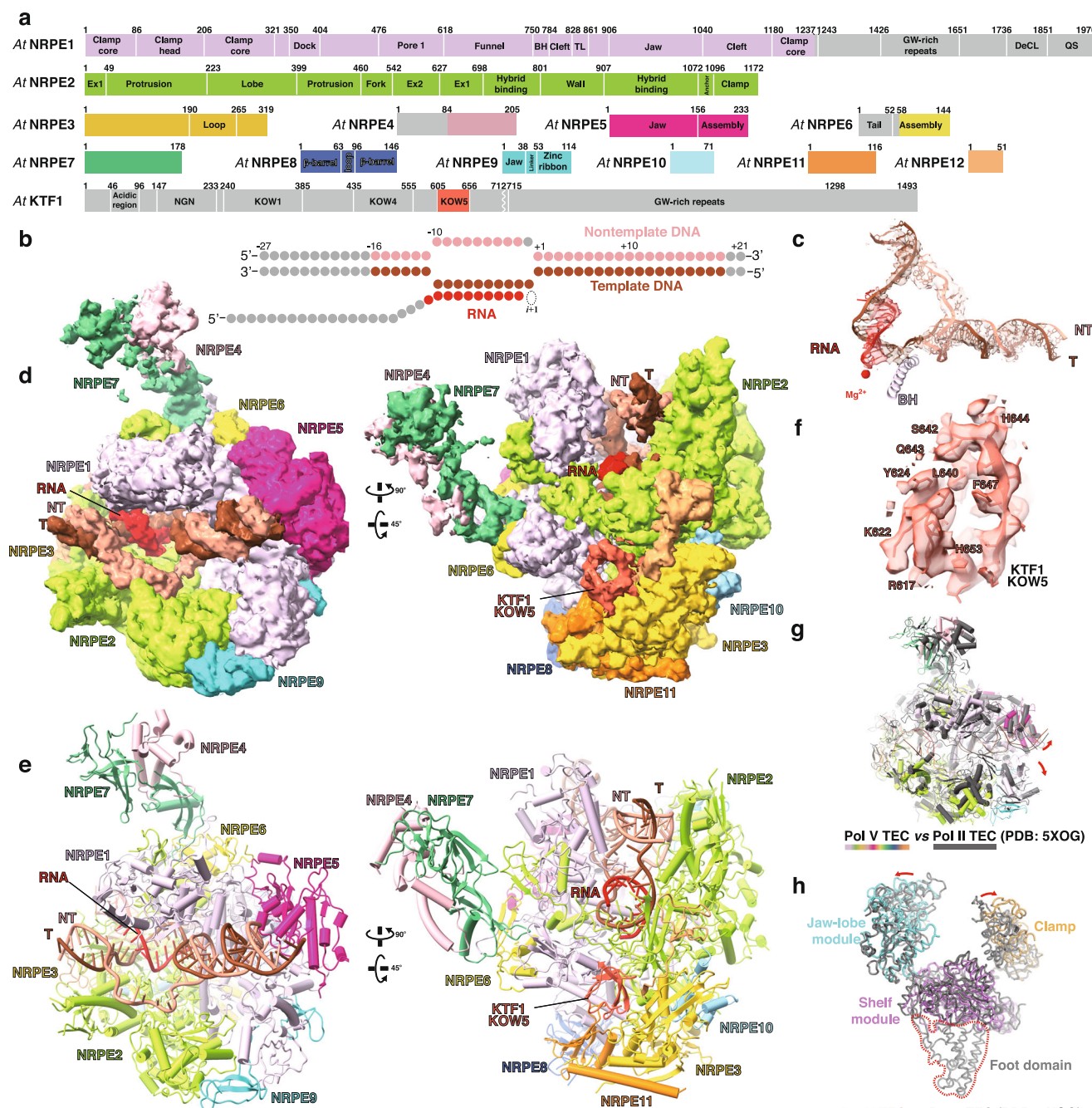

**Fig. 1 | The overall structure of KTF1-bound Pol V TEC. a** The schematic for Pol V subunits and KTF1. The KTF1 domains other than KOW5 are colored in gray due to absence of cryo-EM map signal. The break line indicates the truncated KTF1 (1–712) used in this study. **b** The nucleic-acid scaffold used for cryo-EM structure determination. The dashed ellipse represents the empty 'i + 1' site for binding of the incoming NTP. **c** The 3.2 Å cryo-EM map (map 2) and model for the nucleic-acid scaffold. Red dot indicates the catalytic $Mg^{2+}$ of Pol V. T, template DNA; NT, non-template DNA. **d** The 4.3 Å cryo-EM map (map 1) of two orientations for the KTF1-bound Pol V TEC. **e** Structural model of the KTF1-bound Pol V TEC. **f** The 3.0 Å cryo-EM map (map 3) and model for the KOW5 domain of KTF1. **g** The structural superimposition of the Pol II TEC (PDB: 5XOG) and the Pol V TEC. The red arrows indicate the outward rotation of the clamp and lobe domains of Pol V compared with Pol II. **h** The comparision of the DNA main cleft between the Pol II TEC and the Pol V TEC. The red arrows indicate the outward rotation of the clamp and jaw-lobe domains of Pol V compared to Pol II. The red dash depicts the foot domain that is absence in Pol V.

backbones of the RNA-DNA hybrid make polar and van der Waals interactions with residues in the active-site main cleft, most of which are conserved in Pol II, Pol IV, and Pol V, including NRPE1 residues R325, P415, P416, D451, H456, D453, L772, A773, T777, K780, and NRP(D/E)2 residues K215, R240, F344, R454, S457, N477, N527, K721, Q724, Q725, R806, K921, and H1057 (Fig. 2a). The rudder and lid loops of Pol V are disordered in the Pol V TEC structure, similar to that in Pol IV TEC[18], in contrast to the extensive interactions made by

the two elements with the RNA-DNA hybrid in the structure of Pol II TEC[45,46,49] (Fig. 2c). The disordered conformation of the rudder and lid loops likely results from either the open clamp conformation of Pol V TEC or sequence variations of the two loops compared to Pol II (Supplementary Fig. 3a, b). In summary, the Pol V TEC structure show that the active-site cleft of Pol V retains similar interactions with the RNA-DNA hybrid as Pol II, confering its capacity of RNA extension and DNA translocation (Supplementary Fig. 1d),

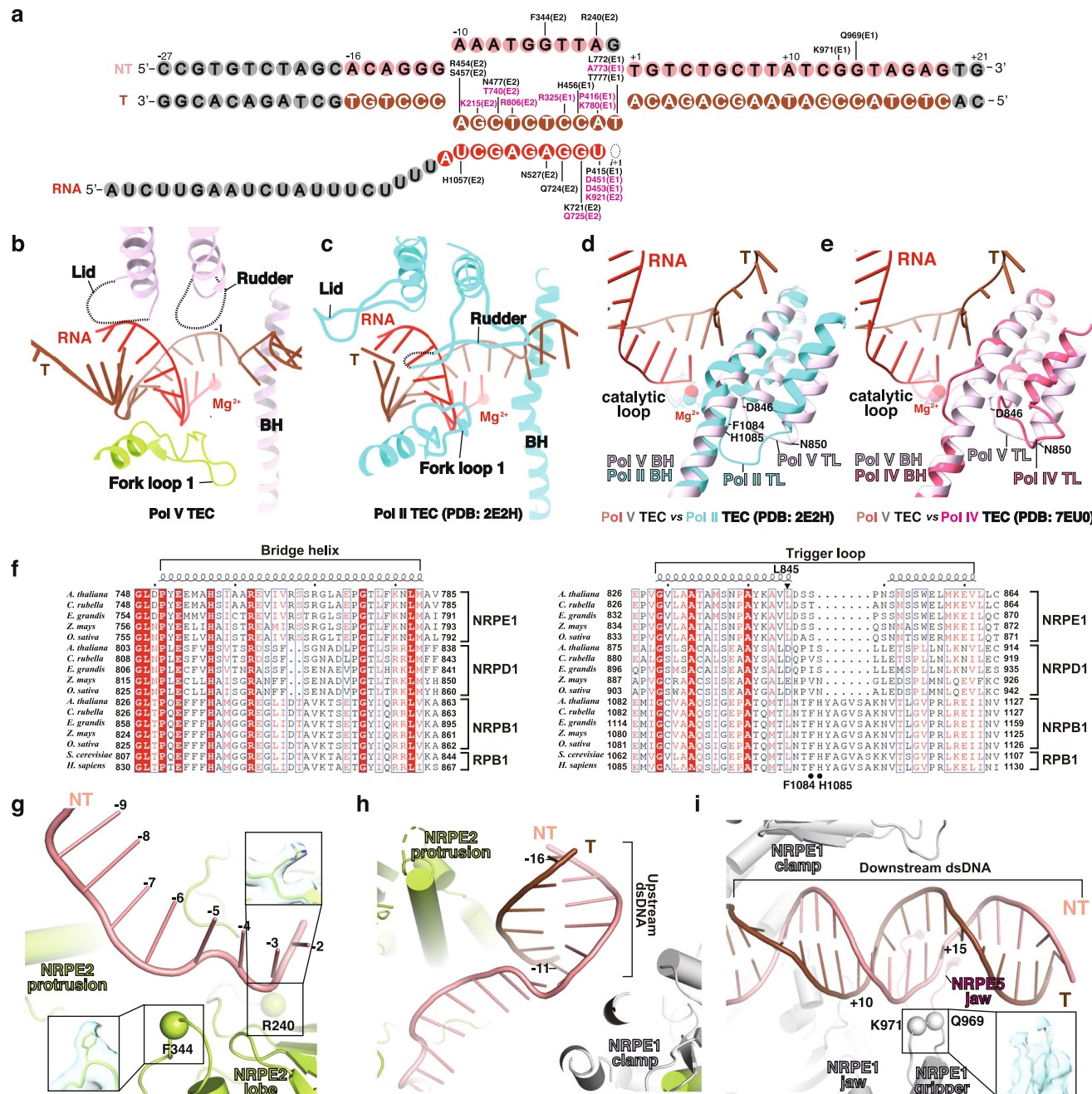

**Fig. 2 | The interactions between Pol V and the RNA-DNA scaffold in the KTF1-bound Pol V TEC. a** The schematic of interaction between Pol V and the nucleic-acid scaffold. Pol V residues making interactions with the DNA and RNA are labeled. The residues conserved between Pol II and Pol V are colored in pink. The nucleotides colored in gray are disordered in the structure. **b** The cartoon presentation of interactions between Pol V and the RNA-DNA hybrid and **c** the interactions between Pol II and the RNA-DNA hybrid in the structure of Pol II TEC (PDB: 2E2H)[52]. The dashes indicate disordered regions. **d** The superimposition of the bridge helix, catalytic loop, and trigger loop between the Pol II TEC (PDB: 2E2H) and the Pol V

TEC (this study) and **e** between the Pol IV TEC (PDB: 7EU0)[18] and the Pol V TEC (this study). **f** The sequence alignment of the bridge helix and trigger loop of Pol II, Pol IV, and Pol V. The residue L845 of *A. thaliana* Pol V NRPE1 is indicated by black filled triangle and the residues F1084 and H1085 of *S. cerevisiae* Pol II RPB1 is indicated by black filled circles. **g** The interactions between non-template ssDNA and Pol V, **h** between upstream dsDNA and Pol V, and **i** between downstream dsDNA and Pol V. The spheres indicate the Cα atoms of polar residues making potential interactions with the phoaphate backbones of DNA.

consistent with the finding the catalytic activity is essential for the function of Pol V in RdDM pathway[16].

The Pol V TEC structure reveals the conformation of the key structural motifs in the active site, namely bridge helix and trigger loop. The 3′-OH moiety of the RNA nucleotide at the 'i' site makes coordination bond with $Mg^{2+}$(I) that is coordinated by the conserved catalytic loop (Fig. 2d). The bridge helix of Pol V-TEC adopts a straight conformation like that of Pol II, in sharp contrast to the bent

conformation of Pol IV bridge helix that contains a two-residue deletion at the middle region (Fig. 2d, e)[18,36]. Sequence analysis shows that bridge helix residues are conserved between Pol II and Pol V but vary significantly between Pol II and Pol IV (Fig. 2f and Supplementary Fig. 3c). The Pol II-like bridge helix of Pol V likely explains its higher NTP incorporation efficiency and processivity than that of Pol IV[36]. In the Pol V TEC structure, the two trigger helices adopt folded helical conformation except for the five residues at the tip loop region

(residues 846–850; Fig. 2d). Intriguingly, Pol V has much shorter trigger loop than Pol II (Fig. 2d, f, and Supplementary Fig. 3d). Sequence analysis suggests that the trigger loop of Pol V keeps the conserved leucine residue (L845 of NRPE1) for contacting the base moiety of the incoming NTP but loses residues for contacting the triphosphate group of the incoming NTP (F1084 and H1085 of yeast Pol II RPB1 subunit) (Fig. 2f and Supplementary Fig. 3d). We infer the shorter trigger loop might explain the lower catalytic activity of Pol V compared with Pol II[36]. In short, the sequence and structural comparison of the key structure motifs (bridge helix and trigger loop) in the active site of Pol V, Pol IV, and Pol II provides structural explanation for the differences of transcription activities of the three RNA polymerases[36].

The Pol V TEC structure also reveals path for the single-stranded nontemplate DNA of the transcription bubble, the upstream dsDNA, and the downstream dsDNA (Fig. 1c, d). The nontemplate ssDNA is restrained in the upper level of the DNA main cleft, where NRP(D/E)2 residues F344 and R240 make interactions with the phosphate backbones of the nontemplate ssDNA (Fig. 2g). In the upstream dsDNA channel, the upstream dsDNA is loosely restrained by the protrusion domain and coiled-coil region of the clamp domain (Fig. 2h), and thereby has much weaker map signal compared with that of the downstream dsDNA (Fig. 1c). In the downstream dsDNA channel, the dsDNA is supported by the NRPE5 jaw domain on the bottom and loosely restrained by the clamp head and NRPE1 jaw domain at the channel sidewalls (Fig. 2i). Residues Q969 and K971 of the NRPE1 gripper likely make polar interaction with the phosphate backbone of the dsDNA (Fig. 2i).

Our structure reveals that the DNA main cleft of KTF1-bound Pol V TEC adopts a much more opened conformation compared that of Pol II TEC (Fig. 1g, h, and Supplementary Fig. 4). The wide-open conformation is not attributed to KTF1 binding, as the recently reported structure of Pol V TEC without KTF1 also shows a wide-open DNA main cleft[50]. Structural comparison between our Pol V TEC and Pol II TEC shows that the clamp domain of Pol V-NRPE1 subunit makes much less interactions with the RNA-DNA hybrid and the downstream dsDNA compared with that of Pol II (Supplementary Fig. 4). The clamp domain of Pol V exhibits disordered lid and rudder loops, and therefore loses interaction with the RNA-DNA hybrid. Moreover, the clamp head domain of Pol V doesn't contain the two DNA grippers of Pol II that interact with downstream dsDNA. We infer that DNA binding is not able to induce clamp closure of Pol V due to the loss of above interactions, resulting in a wide-open DNA main cleft in the Pol V TEC structure. It is unclear how the open conformation of Pol V is related to the function of Pol V, but it might partially account for the slower RNA elongation rate of Pol V creating time windows for AGO4/6 recruitment during Pol V transcription elongation[36].

### The structural features of Pol V-specific subunits

The Pol V-TEC structure reveals the interface between NRPE1 and NRPE5, the two Pol V-specific subunits in *Arabidopsis thaliana*[48]. NRPE5 is exclusively associated with NRPE1 (Fig. 1d and Supplementary Fig. 5c). The structural analysis of Pol V TEC shows that NRPE5-NRPE1 interaction includes ~44 residues of NRPE5 and ~54 residues of NRPE1 constituting a 2109 Å$^2$ otherwise solvent-exposed interface (Fig. 1d, and Supplementary Figs. 5c–e, 6). The structural comparison and sequence analysis between Pol V TEC and Pol II TEC suggest that 17 out of 44 NRPE5 residues and 34 out of 54 NRPE1 residues are unique to the NRPE5-NRPE1 interface, conferring the subunit specificity (Supplementary Fig. 6). Notably, the structure reveals direct interaction between NRPE1 subunit and residues 14–26 of the NRPE5-specific N-terminal extension (Supplementary Fig. 5c–e). Previous report showed overexpression of an NRPE5a derivative with deletion of the N-terminal extension failed to rescue *nrpe5-1* phenotype, highlighting the importance of such interaction[48]. In short, we show that NRPE1-NRPE5 interaction and RPB1-RPB5 interaction use the same surface patches,

but the variation on the interface residues confers the subunit specificity of Pol V.

The cryo-EM map shows that the NRPE7-NRP(D/E)4 stalk associates with Pol V core in a similar manner as the RPB7/4 stalk associates with Pol II core (Fig. 1d, and Supplementary Fig. 5f, g). Although the current map resolution does not allow assignment of interface residues between NRPE1 and NRPE7, sequence alignment shows that the RPB1-RPB7 interface residues are not conserved in NRPE1 and NRPE7, likely accounting for the NRPE7 subunit specificity of Pol V (Supplementary Fig. 7).

### Pol V loses interface for the general transcription factors of Pol IV and Pol II

The Pol V TEC structure explains why Pol V does not interact with RDR2, the backtracking factor of Pol IV. RDR2 makes interactions mainly with the stem and funnel helices of NRPD1 and the zinc ribbon domain of NRP(B/D/E)9[18]. Structural comparison shows that Pol V does not contain the 42-residue Pol IV-specific insertion at the tip of funnel helices, which interacts with NRP(B/D/E)9 zinc ribbon of Pol IV and creates the major RDR2 contact surface (Fig. 3a). Moreover, the funnel helix residues of NRPD1 used for anchoring RDR2 are not conserved in Pol V (Supplementary Fig. 8). In short, Pol V has a secondary channel different from Pol IV to prevent association with RDR2.

Our structure predicts that Pol V does not interact with the elongation factor (TFIIS) and the initiation factors (TFIIB, TFIIE, and TFIIF) of Pol II. In the secondary channel, the zinc ribbon domain of NRP(B/D/E)9 occupies the entry route of the TFIIS interdomain linker (Fig. 3b). Therefore, unless there is a large conformational change, Pol V is not able to recruit TFIIS to rescue arrested Pol V TEC. Detailed structural comparison and sequence alignment between Pol V and Pol II show that, similar to Pol IV, Pol V lacks anchoring surfaces for Pol II initiation factors, including TFIIB, TFIIF, TFIIE (Fig. 3c–e, and Supplementary Figs. 9 and 10). The result supports that Pol V does not recognize Pol II promoters and utilizes a distinct mechanism for transcription initiation[29,38].

### The KOW5 domain of KTF1 locates at the RNA exit channel of Pol V

Our cryo-EM map of KTF1-bound Pol V TEC reveals unambiguous signal for the KOW5 domain of KTF1 (Fig. 1d, f), although the KTF1-SPT4 heterodimer was included during complex reconstitution. No cryo-EM map signal is observed for NGN domain of KTF1 and SPT4 near the upstream dsDNA channel (Fig. 1d), where SPT5-NGN and SPT4 locate in the Pol II TEC (Supplementary Fig. 5b)[45,46]. The structure shows that KOW5 binds near the RNA exit channel, where it makes interactions with the C-terminal β strand of NRP(A/B/C/D/E)12 subunit, the wall (or flap) domain of the NRP(D/E)2 subunit, and zinc loop of NRP(B/D/E)3 subunit (Fig. 4a). KTF1-KOW5 domain interacts with Pol V mainly through polar interactions made by R615, R617, K622, Y624, R627, K639, S642, Q643, and H644 (Fig. 4b, c). The residues on the reciprocal interface on Pol V includes H51, E91, N94, D96, and R170 of NRPE(B/D/E)3, residues Q47 and E49 of NRP(A/B/C/D/E)12 (Fig. 4b, c), and residues S841, N853, and S855 of NRP(D/E)2 (Fig. 4b). In short, the KTF1-KOW5 domain interacts with NRP(B/D/E)3, NRP(A/B/C/D/E)12, and NRP(D/E)2 subunits mainly through polar interactions.

Our structure reveals slight difference on the interface of KTF1-KOW5 domain and Pol V compared with that of SPT5-KOW5 domain and Pol II. On the NRP(D/E)2-KTF1 interface, K639 of KTF1 KOW5 replaces residue E781 of SPT5 KOW5 at the corresponding position and makes a H-bond with S855 of NRP(D/E)2; residues N853 and S841 of NRP(D/E)2 also replace RPB2 residues R904 and K892, respectively, which make interactions with E781 of SPT5 KOW5 (Fig. 4e and Supplementary Fig. 12b). Moreover, sequence alignment shows that Arabidopsis SPT5 has two-residue deletion (corresponding to KTF1 S642 and Q643) in the β3-β4 loop that participates into the interactions with

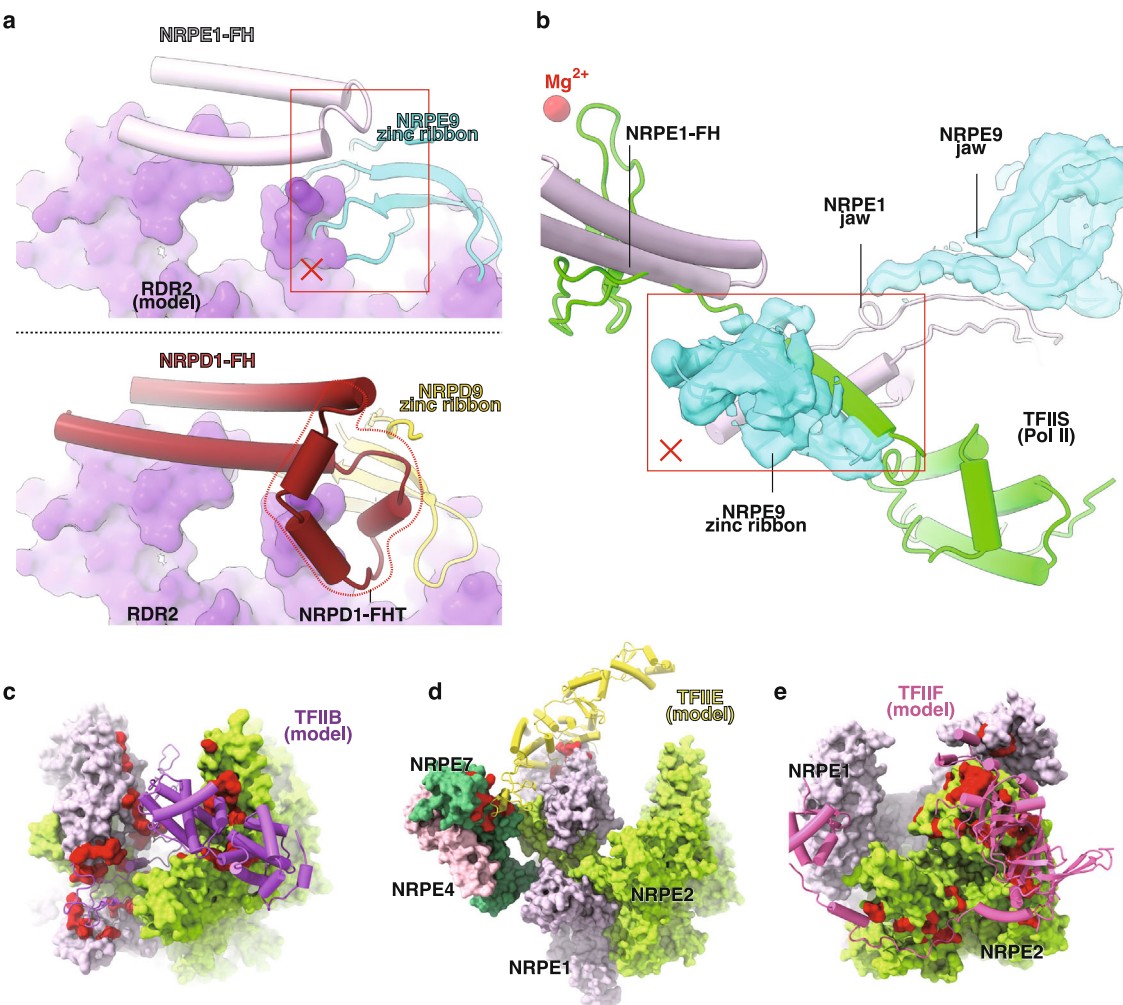

**Fig. 3 | Pol V does not interact with the initiation and elongation factors of Pol II and Pol IV. a** Pol V is not compatible with RDR2. The upper panel shows clashes between the modeled RDR2 (purple surface; based on superimposition of the Pol IV-RDR2 (PDB: 7EU0) and Pol V TEC structures) and Pol V. The red rectangle indicates clashes between the modeled RDR2 and the zinc ribbon domain of NRP(B/D/E)9. The lower panel shows the interaction between RDR2 and Pol IV in the Pol IV-RDR2 structure (PDB: 7EU0)[18]. The Pol IV-specific insertion of at the funnel helix tip (NRPD1-FHT) is highlighted by dashes. **b** Pol V is incompatible with TFIIS binding. The red rectangle indicates clashes between the modeled TFIIS (based on superimposition of the Pol II-TFIIS (PDB: 1Y1V) and Pol V structures)[53] and the zinc ribbon domain of NRP(B/D/E)9 shown in cryo-EM map and ribbon. **c** The surface patches of Pol V is incapable of interaction with TFIIB, **d** TFIIE, and **e** TFIIF. TFIIB, TFIIE, and TFIIF are modeled on the Pol V TEC structure based on the superimposition between human Pol II PIC (PDB: 5IYC)[54] and Pol V TEC. The red surface patches highlight non-conserved residues of Pol V on the corresponding initiation factors-contact surfaces of Pol II.

the NRP(D/E)2 subunit of Pol V (Supplementary Figs. 11 and 12), suggesting less preferred interaction between SPT5 and Pol V. Except for the above differences, similar residues are used by KTF1 KOW5 and SPT5 KOW5 to interact with NRP(B/D/E)3 and NRP(A/B/C/D/E)12 subunits that are shared between Pol II and Pol IV (Supplementary Fig. 12c). In summary, our structure shows detailed interface between Pol V and KTF1-KOW5 domain that partially accounts for the specificity of AGO4/6-recuting factor of Pol V.

## Discussion

In summary, the structure of KTF1-bound Pol V TEC shows that Pol V retains a Pol II-like active center for its basic activity of RNA extension and DNA translocation. The sequence variation and conformational differences of the structural motifs on Pol V-NRPE1 subunit, including the trigger loop, lid loop, and rudder loop, explains the inferior ability of extending RNA compared with Pol II. A recent report of cryo-EM structures of *Brassica oleracea* Pol V TEC proposed a possible role in Pol V chromatin retention of the interaction between Pol V and the nontemplate ssDNA as well as the ss/dsDNA junction of the transcription bubble[50]. However, our cryo-EM

map of KTF1-bound Pol V TEC displays poor signals at above regions and therefore the possible role is not discussed. Moreover, the blockage of entry channel of TFIIS by NRP(B/D/E)9 suggests Pol V transcription elongation is susceptible to arresting compared with Pol II. These features together suggest that Pol V may pause frequently during elongation to coordinate with AGO4/6 recruitment. A recent reported cryo-EM structure of Pol V TEC exhibits similar conformations of the structural motifs at the active-site cleft of Pol V, supporting our conclusion.

The structure of Pol V TEC reveals specified interface between NRPE1 and NRPE5 and predicts the existence of specified interface between NRPE1 and NRPE7, explaining the unique subunits NRPE1, NRPE5, and NRPE7 of Pol V. Moreover, the structure of Pol V TEC shows the corresponding surface patches, which are used for anchoring transcription initiation factors of Pol II, contain sequence mutation, insertion, or deletion in Pol V, explaining Pol V does not initiate transcription from Pol II promoters[29,38]. Finally, the Pol V TEC structure reveals the difference of structural motifs at the secondary channels between Pol IV and Pol V, explaining the absence of RDR2 association with Pol V.

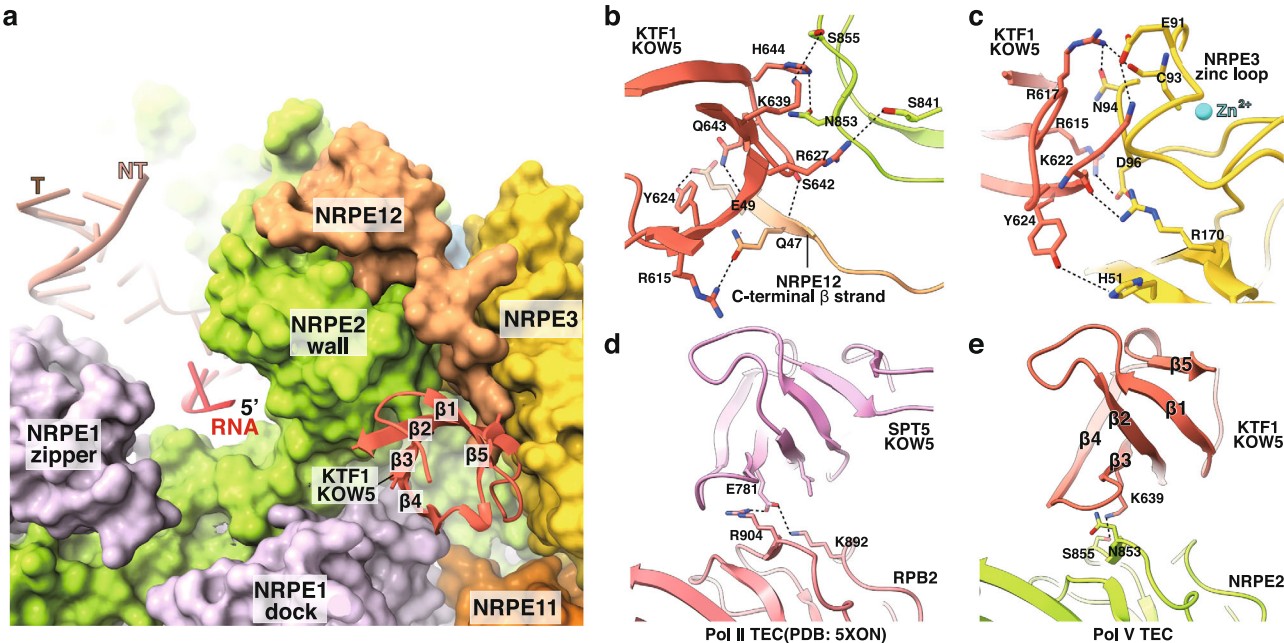

**Fig. 4 | The KOW5 domain of KTF1 locates at the RNA exit channel. a** The overall structure showing that the KOW5 domain of KTF1 in cartoon bound near the RNA exit channel in surface. The β strands on KTF1 KOW5 are numbered from the N to C terminus. **b** The detailed interactions between KTF1 KOW5 and the NRP(D/E)2 and NRP(A/B/C/D/E)12 subunits of Pol V. **c** The detailed interactions between KTF1 KOW5 and the NRP(A/B/C/D/E)3 subunit of Pol V. **d** The detailed interactions between SPT5 KOW5 and the RPB2 subunit of yeast Pol II (PDB: 5XON)[46]. **e** The detailed interactions between KTF1 KOW5 and the NRP (D/E)2 subunit of Pol V. Dashes indicate H-bond interactions.

Our structure reveals the interaction between KTF1 KOW5 and Pol V. KTF1-KOW5 domain binds near the RNA exit channel of Pol V, analogous to the interaction between SPT5-KOW5 domain and Pol II in the structure of Pol II TEC. Our structure doesn't show interaction of SPT4 and other domains of KTF1 with Pol V. The absence of interaction between Pol V and KTF1 NGN is likely attributed to the wide-open main DNA cleft that prevents interaction of KTF1 NGN with the coiled-coil domain of Pol V. Based on our structure of KTF1-bound Pol V TEC, we proposed a more refined model of Pol V-centered DNA methylation complex[23,44] (Fig. 5). The structure shows that Pol V accommodates the upstream dsDNA, downstream dsDNA, and the RNA-DNA hybrid in a similar manner as Pol II. Moreover, our structure and in vitro transcription results suggest that Pol V can extend RNA primer of a three-stranded RNA-DNA scaffold. Therefore, we speculate that Pol V maintains a ~10-bp transcription bubble during its elongation as Pol II. Our structure indicates that the C-terminal GW motifs of both KTF1 and NRPE1 point to the upstream direction of Pol V, allowing the recruitment of the siRNA-loaded AGO4/6 to the upstream DNA region of Pol V TEC. It is still unclear how DRM2 is subsequently recruited to the complex and how the interaction evokes its methyltransferase activity. A structure of the complete Pol V transcription-coupled DNA methylation complex is required to decipher the mechanism in the future.

In summary, our structure of KTF1-bound Pol V TEC complex provides the structural basis for understanding Pol V transcription elongation. Moreover, our structure provides a start point to study the Pol V elongation-coupled DNA methylation.

## Methods

### Plasmids

The plasmid pLZ009 -GFP-NRPE1 was constructed by inserting the DNA sequence encoding *Arabidopsis thaliana* NRPE1 (At2g40030) by a homogenous recombination method (pEASY®-Basic Seamless Cloning and Assembly Kit, Transgen Biotech, Inc) into a modified pLZ009 vector (pLZ009-GFP), which contains the *CaMV35S* promoter, the *NOS* terminator, and a GFP-expressing cassette consisting of the *UBQ10* promoter,

the coding sequence of GFP with the N7 nuclear localization signal at its N-terminus, and the *UBQ10* terminator. The plasmid pETDuet1-KTF1-SPT4 was constructed by inserting the coding region of KTF1 (residues 1–712; At5g04290) and SPT4 (At5g63670) into modified pETDuet1 vector (Novagen), which contains an 6xHis tagged SUMOstar[51].

### Generation of *A. thaliana* T87-Pol V stable cell line

The *Arabidopsis thaliana* T87 cell line stably expressing GFP and 3xFLAG-10xHis-NRPE1 (T87-Pol V) was constructed as in ref. 18. Briefly, T87 cells were mixed with *A. tumefaciens* cells carrying pLZ009-GFP-NRPE1 for 2 days and subsequently plated on agar plates (JPL3 solid medium supplemented with 250 µg/mL carbenicillin and 30 µg/mL kanamycin). The kanamycin-resistant calli were picked and passaged every 2 weeks for three times. The calli were subsequently collected for detecting the expression of 3xFLAG-10xHis-NRPE1 by western blot with an anti-flag antibody and for examining the ratio of GFP-positive cells under a fluorescence microscopy. The T87-Pol V callie, which exhibit high expression level of 3xFLAG-10xHis-NRPE1 and 100% GFP-positive cells, were cultured in the JPL3 medium under continuous illumination at 22 °C on rotary shaker at 120 rpm.

### Protein expression and purification

The epitope-tagged *Arabidopsis thaliana* Pol V was purified essentially as described in ref. 18. In brief, T87-Pol V cells (~5 kg a batch) were harvested and lysed in liquid nitrogen using SKSI Tissue Lyser (BiHeng) and then homogenized in 5 L lysis buffer (100 mM Tris-HCl, pH 7.7, 300 mM NaCl, 10% glycerol, 2 mM DTT, 0.1 mM PMSF, and protease inhibitor cocktail (APExBIO,Houston,USA). The homogenized material was centrifuged at 15000 g for 60 min. The supernatant was further filtered through two layers of Miracloth (Calbiochem) and precipitated by ammonium sulfate (350 g/L) for 5 h. After centrifugation at 15000 g for 60 min, the pellet was dissolved in TGED buffer (50 mM Tris-HCl, pH 7.7, 10 mM (NH$_4$)$_2$SO$_4$, 10% glycerol, 2 mM DTT, and 1 mM EDTA) followed by centrifugation at 15000 g for 60 min. The supernatant was incubated with anti-FLAG affinity resin (GenScript) on a rotator for 2 h,

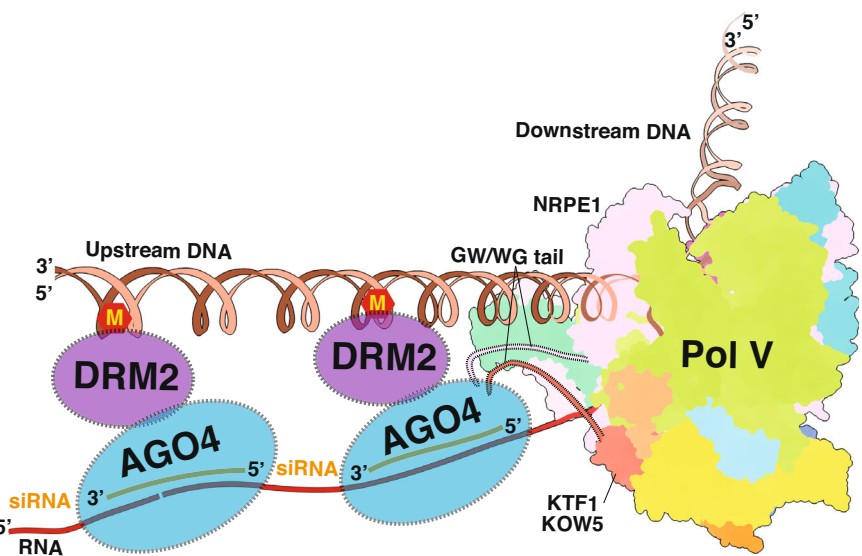

**Fig. 5 | The model of the Pol V transcription elongation-coupled DNA methylation.** The structure of KTF1-bound Pol V TEC indicates that the C-terminal GW motifs of both KTF1 and NRPE1 point to the upstream direction of Pol V, allowing recruitment of siRNA-loaded AGO4/6 in the upstream of Pol V transcription. 'M' stands the methyl modification on DNA.

and the resulting material was loaded on a gravity column. The resin was washed with 20-column volumes of buffer A (50 mM Tris-HCl, pH 7.7, 150 mM NaCl, 5% glycerol, 5 mM $MgCl_2$, 1 mM DTT), followed by Pol V elution with 5 column volumes of buffer A containing 500 µg/mL FLAG peptide. The elute was loaded onto a Capto Q HiRes column (Cytiva) and Pol V was eluted using a linear gradient from 0.1 M to 0.8 M NaCl in buffer A. The fractions containing target proteins were collected, concentrated and applied to a Superose 6 10/300 GL column (Cytiva) equilibrated in 20 mM HEPES pH 7.5, 100 mM KCl, 5 mM $MgCl_2$, 2 mM DTT. The fractions containing target proteins were collected, concentrated to ~1.0 mg/mL and stored at −80 °C.

The KTF1-SPT4 heterodimer was over-expressed from *E. coli* strain Rosetta (DE3) (Novo protein, Inc.) containing pETDuet1-KTF1-SPT4 at 16 °C under 0.5 mM IPTG induction for overnight. The cells were harvested and lysed in lysis buffer (50 mM Tris-HCl, pH 7.7, 300 mM NaCl, 10% glycerol, 0.1 mM PMSF, and 5 mM β-mercaptoethanol) using an Avestin EmulsiFlex-C3 cell disrupter (Avestin, Inc.). The supernatant was incubated with Ni-NTA resin (Smart-Lifesciences, Inc) on a rotator for 1 h and loaded on a gravity. The KTF1/SPT4 heterocomplex was released from the resin through on-column cleavage by the SUMOstar protease. The protein sample was loaded onto a Heparin column (Cytiva), followed by elution using a linear gradient from 0.1 M to 0.8 M NaCl in buffer A. The fractions containing target proteins were collected, concentrated and loaded onto a Superose 6 10/300 GL column (Cytiva) equilibrated in 20 mM HEPES pH 7.5, 100 mM KCl, 5 mM $MgCl_2$, 2 mM DTT. The fractions containing the protein complex were collected, concentrated to ~2.0 mg/mL and stored at −80 °C.

**Mass spectrometry**

The protein sample (200 µL) in UA buffer (8 M Urea, 150 mM Tris-HCl, pH 8.0) was mixed with 50 mM IAA (100 µL) in UA buffer and incubated in darkness at room temperature for 30 min. The reaction mixture was loaded into a filter device and centrifuged at 14,000 g for 15 min. The concentrate was diluted in the device with 200 µL of UA buffer and centrifuged again. This step was repeated twice. The concentrate was subsequently diluted with 100 µL of 25 mM $NH_4HCO_3$ and concentrated again. This step was repeated twice. The concentrate was diluted to 40 µL of 25 mM $NH_4HCO_3$ containing 2 µg of Trypsin. The peptide mixture was loaded onto a C18-reversed phase column (15 cm long, 75 µm inner diameter) packed in-house with RP-C18 5 µm resin in buffer A (0.1% Formic acid in HPLC-grade water) and separated with a

linear gradient of buffer B (0.1% Formic acid in 84% acetonitrile) at a flow rate of 250 nl/min over 60 min. MS data of the peptides were acquired on a Q Exactive mass spectrometer coupled to EASY-nLC (Thermo Fisher Scientific) using a data-dependent top 10 methods that dynamically choose the most abundant precursor ions from the survey scan (300–1800 m/z) for HCD fragmentation. Determination of the target value is based on predictive Automatic Gain Control (pAGC). Dynamic exclusion duration was 20 s. Survey scans were acquired at a resolution of 70,000 at m/z 200 and resolution for HCD spectra was set to 17,500 at m/z 200. Normalized collision energy was 27 eV and the underfill ratio was defined as 0.1%, which specifies the minimum percentage of the target value likely to be reached at maximum fill time. The instrument was run with peptide recognition mode enabled. The experiment was performed once.

MS/MS spectra were searched using MASCOT engine (Matrix Science, London, UK; version 2.2) against Arabidopsis sequence database (download in March 2021; 136838 sequences). For protein identification, the following options were used. Peptide mass tolerance=20 ppm, MS/MS tolerance=0.1 Da, Enzyme=Trypsin, Missed cleavage=2, Fixed modification: Carbamidomethyl (C), Variable modification: Oxidation (M), ion score > 20 at peptide and protein level.

**Nucleic-acid scaffold preparation**

The nucleic-acid scaffold used for cryo-EM structure determination contains a template DNA (5′-CACTCTACCGATAAGCAGACATACCT CTCGACCCTGTGCTAGACACGG–3′), a nontemplate DNA (5′-CCG TGTCTAGCACAGGGAAATGGTTAGTGTCTGCTTATCGGTAGAGTG-3′), and an RNA (5′-AUCUUGAAUCUAUUUCUUUUAUCGAGAGGU-3′). The RNA, template DNA, and nontemplate DNA were incubated in a 1:1.1:1.2 molar ratio and annealed in 20 mM Tris-HCl, pH 8.0, 200 mM NaCl (95 °C for 5 min followed by a 2 °C/min cooling procedure to a final temperature of 22 °C in a thermo cycler).

The nucleic-acid scaffold used for in vitro transcription assays contains a template DNA (5′-CACTCTACCGATAAGCAGACATAC CTCTCGATCCTGTGCTAGACACGG-3′), a nontemplate DNA (5′-CCGTGTCTAGCACAGGTAAATGGTTTGTGTCTGCTTATCGGTAGAG TG-3′) and an RNA (5′-ACAGAUCGUGUCCAUCGAGAGGU-3′). The RNA and template DNA were incubated in a 1:1.1 molar ratio and annealed in 20 mM Tris-HCl, pH 8.0, 200 mM NaCl (95 °C for 5 min followed by a 2 °C/min cooling procedure to a final temperature of 22 °C in a thermal cycler).

## In vitro transcription assays

To measure RNA extension activity of Pol V, the Pol V elongation complex was prepared by incubating 200 nM (final concentration) RNA-template DNA scaffold and 100 nM (final concentration) *A. thaliana* Pol V for 30 min followed by addition of 400 nM non-template DNA and incubation for additional 30 min at room temperature in 20 μL transcription buffer (20 mM HEPES, pH 7.5, 100 mM NaCl, 5 mM $MgCl_2$, 1 mM DTT). The transcription reactions were initiated by addition of a NTP mix (final concentration: 100 μM each of GTP, CTP and ATP, 1 μM UTP) and 4 μCi of [$\alpha$-$^{32}$P]UTP (3000 Ci/mmol; Perkin Elmer), allowed to proceed for 1 h at room temperature, and then stopped by addition 5 μL stop buffer (8 M urea, 20 mM EDTA, 0.025% xylene cyanol and 0.025% bromophenol blue). The samples were boiled for 5 min, and immediately cooled in ice for 5 min.

## Cryo-EM sample preparation

The KTF1-SPT4-bound Pol V TEC was assembled by sequential addition of Pol V, nucleic-acid scaffold, and KTF1/SPT4 in a 1:1.2:3 molar ratio in 20 mM HEPES, pH 7.5, 100 mM NaCl, 5 mM $MgCl_2$, 1 mM DTT. The mixture was subsequently applied to a Superose 6 Increase 10/300 GL column (Cytiva) and KTF1-SPT4-bound Pol V TEC was eluted in 20 mM HEPES, pH 7.5, 100 mM KCl, 5 mM $MgCl_2$, 2 mM DTT. Fractions containing the complex were collected and concentrated to ~2.4 mg/mL.

Quantifoil R1.2/1.3 300 mesh Cu holey carbon grids were glow-discharged for 30 s with a mixture of $H_2$ and $O_2$ (1:2) using a glow-discharge cleaning system (Solarus II Plasma Cleaner, Gatan, Inc). The KTF1-SPT4-bound Pol V TEC was mixed with 0.005% Tween-20 (Sigma; v/v; final concentration) and 3 μL mixture was applied on the grids in the chamber of Vitrobot Mark IV (FEI; 95% humidity; 22 °C), blotted for 3 s with blot force −2, and vitrified by plunging it into liquid ethane.

## Cryo-EM data collection and processing

Cryo-EM data were collected using EPU in the super-resolution mode on a 300 kV Titan Krios (FEI) equipped with a Gatan K3 Summit direct electron detector (pixel size 1.10 Å/pixel). A total of 6811 images were recorded using the counting mode (exposure, 2.67 s per 40-frame movie; dose rate, 22.5 electrons/pixel/s; defocus, −1.2 to −2.2 μm). Frames of individual movies were aligned using MotionCor2 and contrast-transfer-function estimations were performed using CTFFIND4. Image processing was performed with RELION 3.1. A total of 3,436,012 particles were auto-picked and extracted (box size, 256 pixels; pixel size, 1.1 Å) by using 2D classified templates generated from the particles that are used for calculating the final 6.7 Å final 3D model of a preliminary dataset. The particles were subsequently subjected to 2D classification (N = 30, iterations=25), resulting 2D classes with distinct shape of Pol V containing 503,881 particles. The particles were further subjected to two-round 3D classifications using a 40 Å low-pass-filtered 6.7 Å structural model calculated from the preliminary dataset as the initial model. The single particles (273,007) of the top two classes were subjected to auto refinement, CTF refinement, particle polishing, and postprocess, resulting in a 3.2 Å map (map 2). The same set of single particles were imported into cryoSPARC and subjected to non-uniform refinement, resulting in a 3.13 Å map. Two masks were created for subsequent steps of particle subtraction and local refinement to obtain map 3 and map 4. The mask for map 3 includes the core module of Pol V and KTF KOW5 domain, while the mask for map 4 includes the NRPE5 and NRP(B/D/E)9 subunits. Map 3 and map 4 were refined to a resolution of 2.97 Å and 3.70 Å, respectively. A lower resolution map (4.27 Å; map 1) showing more complete signal for the clamp domain was calculated from 30359 single particles. The AlphaFold2-predicted structure models of Pol V and KTF1 were used as the start model for model building. The model was fit into the cryo-EM maps. Iterative cycles of model building in Coot (Ramachandran, trans peptide, planar peptide restraints applied) and refinement in Phenix were subsequently performed.

## Reporting summary

Further information on research design is available in the Nature Portfolio Reporting Summary linked to this article.

## Data availability

The data that support this study are available from the corresponding authors upon reasonable request. The cryo-EM density map has been deposited in the EM Databank under accession code EMD-35086 (KTF1-bound Pol V TEC). The atomic coordinates have been deposited in the RCSB Protein Data Bank (PDB) under accession codes 8HYJ. Previously published structures used in this study can be accessed via accession codes 5XON, 2E2H, 1Y1V, 5IYC, 7EU0 and 5XOG. The mass spectrometry proteomics data have been deposited in ProteomeXchange Consortium (http://proteomecentral.proteomexchange.org) via PRIDE repository with the dataset identifier accession number PXD041996. Source data are provided with this paper.

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

## Acknowledgements

The work is supported by National Key Research and Development Program of China 2020YFA0907902 (Y.Z.), Basic Research Zone Program of Shanghai JCYJ-SHFY-2022-012 (Y.Z.). We thank Dr. Liangliang Kong, Dr. Fangfang Wang, Dr. Guangyi Li, and Dr. Jialin Duan at the cryo-EM center of NFPS in Shanghai, Weida Liu and Prof. Zhenguo Chen at the cryo-EM center of Fudan University for the supports on cryo-EM data collection and data analyses.

## Author contributions

H.-W.Z., K.H., and Z.-X.G. constructed the Pol V-T87 stable cell line and purified epitope-tagged Pol V. Z.-X.G. and H.-W.Z. purified KTF1 and SPT4. H.-W.Z., K.H., X.-X.W., W.L. collected the cryo-EM data and solved the cryo-EM structure. Y.X. and Y.Z. assisted in structure determination. J.-W.W., Y.Z. designed experiments. Y.Z. and J.-W.W. wrote the manuscript.

## Competing interests

The authors declare no competing interests.
