## [Peer Review File · Nature Communications]

A cryo-EM structure of KTF1-bound polymerase V transcription elongation complexREVIEWER COMMENTS

Reviewer #1 (Remarks to the Author):

Using cryo-EM single-particle analysis, this important study by Zhang/Huang/Gu et al. shows the first structure of plant-specific nuclear DNA-dependent RNA Polymerase V (Pol V) bound by Kyrpides-Ouzounis-Woese (KOW) domain-containing transcription factor 1 (KTF1; also known as SUPPRESSOR OF TY INSERTION 5-LIKE, SPT5L), together with a nucleic acid scaffold composed of RNA, template DNA, and non-template DNA to mimic the RNAs of an elongation complex. The overall architecture of Pol V with the DNA/RNA scaffold is very similar to that of Pol II. Unique amino acid compositions in catalytic motifs such as the bridge helix and trigger loop may explain characteristics of Pol V transcription activity reported previously by others. The Pol V-specific N-terminal extension of the 5th subunit (NRPE5) contributes to the unique interaction interface with NRPE1 (the largest subunit), explaining the subunit and polymerase-specificity of the interaction. Also, the identification of the KOW5 domain of KTF1 in the cryo-EM map confirms that its binding location is similar to that of SPT5 when interacting with Pol II. A few Pol V-KTF1 specific differences in the interacting amino acids may account for the Pol V specificity of the interaction. The KTF1-KOW5 domain is located near the RNA exit channel, which would place Glycine-Tryptophan (GW)-rich repeats of KTF1 in the vicinity of Pol V nascent transcripts. These GW repeats of KTF1 and the GW repeats of the Pol V C-terminal domain were shown to be genetically redundant by Lagrange and colleagues in 2016 (see citation information below) and serve as so-called AGO hooks for binding AGO4. Moreover, a new paper by Wang et al. (see citation information below) has shown that ARGONAUTE 4 (AGO4) slices and retains its target RNAs in vivo and in vitro, with prior studies showing that Pol V is the source of the target RNAs in vivo. Collectively, these findings, which are not currently cited, together with the finding that AGO4 can physically interact with DOMAINS REARRANGED METHYLTRANSFERASE 2 (DRM2) (Zhong et al., cited as reference 39 in the current manuscript) support the model the authors present at the end of the paper. The new cryo-EM structure provides the structural context for this model and a basis for future studies exploring how additional proteins associate with the complex to bring about DNA methylation.

Lahmy, S., Pontier, D., Bies-Etheve, N., Laudie, M., Feng, S., Jobet, E., Hale, C.J., Cooke, R., Hakimi, M.A., Angelov, D., et al. (2016). Evidence for ARGONAUTE4-DNA interactions in RNA-directed DNA methylation in plants. *Genes Dev* 30, 2565-2570. 10.1101/gad.289553.116.

Wang, F., Huang, H.Y., Huang, J., Singh, J., and Pikaard, C.S. (2023). Enzymatic reactions of AGO4 in RNA-directed DNA methylation: siRNA duplex loading, passenger strand elimination, target RNA slicing, and sliced target retention. *Genes Dev*. 10.1101/gad.350240.122.

Major comments/critiques:

1. The term "transcription elongation complex (TEC)" for this study needs to be clarified. The method for "TEC" assembly is derived from other studies with well-characterized RNAPs such as Pol II. But no RNA synthesis and RNA elongation by nucleotide incorporation is demonstrated for the assembled RNA-DNA scaffold, thus it is not clear that the complex is truly an elongation complex in which the synthetic RNA can be extended.

2. Is KTF1 truly an elongation factor? A study in 2009 by He et al, (An effector of RNA-directed DNA methylation in Arabidopsis is an ARGONAUTE 4- and RNA- binding protein. *Cell* 137: 498-508) identified a KTF1 mutant they called DMS3. They examined Pol V transcript levels at several loci in *dms3/ktf1* mutants and found that Pol V transcript levels actually increased in the mutant, not decreased as expected if DMS3/KTF1 is important for transcript production. Pol V transcript levels are also increased in *ago4* mutants, probably because AGO4 slicing of the transcripts does not occur. So, KTF1 function may only be for AGO4 recruitment, without affecting Pol V transcription at all. The authors should discuss this and perform an experiment to test whether KTF1 has any effect on Pol V transcription elongation in vitro. KTF1 is SPT5-LIKE, not SPT5 itself, so it may not have the same function as SPT5, as the authors assume.

3. A catalytic mutant of NRPE1 should ideally have been used as a control in the transcription assays

to prove that the elongation products in Fig. S1D are made by Pol V and not by a contaminating RNA polymerase (e.g., Pols I, II, and III). Including α -amanitin in the transcription assay would at least suppress Pol II's activity, and this is an easy experiment to include.

4. Page 6 line 12: KTF1/SPT4 was added to "increase stability", but no data showing that KTF5/SPT4 addition indeed improves Pol V stability is presented in the manuscript. A gel filtration profile of the purified Pol V with or without KTF1/SPT4 would be informative.

5. What is the reason for using SPT4-2 (AT5G63670), and not SPT4-1 (AT5G08565)?

6. What is the basis for recombinant KTF1/SPT4 forming a "stable heterodimer"? Are KTF1 and SPT4 stoichiometric in Fig S1?

7. The authors state that KTF1/SPT4 forms a stable heterodimer, but this is based only on their co-purification along with other proteins (Fig. S1C), not a demonstration of a direct interaction. The authors should perform reciprocal co-immunoprecipitation or pull-down experiments to confirm the existence of the KTF1/SPT4 heterodimer. Size-exclusion chromatography (SEC) could also be performed to try to isolate and confirm the KTF1-SPT4 heterodimer.

8. The authors refer to "endogenous" Pol V, but this is a bit confusing given that they have isolated epitope-tagged Pol V by virtue of an engineered epitope tag (FLAG-10xHis) on the largest subunit NRPE1. This recombinant subunit is constitutively overexpressed from a strong, non-native 35S promoter, and the transgene is potentially inserted in the genome multiple times. The impact of skewing subunit abundance by constitutive overexpression of only one of 12 subunits on the Pol V assembly pathway is not known. Since the transgenic line was established in the wild-type background, the T87 cell line should also contain Pol V complexes assembled with native NRPE1 subunit, which is the truly endogenous form of Pol V. Perhaps the authors should refer to their complex as "in vivo assembled" or "epitope-tagged" Pol V.

9. Table S1: As is, the Table S1 title should be "Pol V subunits detected by LC-MS/MS" because it does not describe any of the other proteins that must have been detected in the analysis of the impure sample. There are a great many unidentified proteins in the SDS-PAGE gel shown for the Pol V complex. It would be useful to have the full mass spec report for the proteins present in the fractions provided as a supplemental table.

10. Related to points 9 and 10, does the FLAG tag purified Pol V complex include KTF1 or SPT4? How many replicates did the authors use in the IP-MS experiment, and what is the negative control (e.g. IP of extracts from non-transgenic plants)?

11. page.7 lines 15-17: The phrase "including NRPE1 residues R325, P416, D451, A773, K780, Q869, and NRPE(D)2 residues K215, K244, R420, Q725, T740, R806 (Fig. 2A)" is confusing because P416, D451, A773, Q869, Q725, T740 are not positively charged. Also, in the model, D453(E1), rather than D451(E1), appears to make interaction with i-1 U. K921(E2) also seems to interact with i-1 U but this is not included in Fig. 2A? What are the criteria used for claiming specific amino acid interactions?

12. It is intriguing that the purified Pol V from the transgenic T87 Arabidopsis cells has NRPE5c (AT3G54490) exclusively (according to the LC-MS/MS, Table S1), and not NRPE5a (AT3G57080) that have been detected as the predominant Pol V 5th subunit in other studies from multiple labs*. In this work, the cryo-EM single-particle analysis was able to detect subunits that were not found in LC-MS/MS. Because the structural basis of NRPE5-specific Pol V association is a major part of this manuscript, the identity of the 5th subunit in the reconstruction would need to be clarified further. Examples of these studies are Y Li et al. *Plant Cell* (2018); G Qin et al. *Sci Rep* (2017); JA Law et al. *Curr Biol* (2010); TS Ream et al. *Mol Cell* (2009). This point should be clarified and discussed in the main text, in addition to the legend of Fig. S5.

13. Related to the point above, one wonders if in the cultured cell line T87, how NRPE5a~c expressed? Compared to NRPE5a(AT3G57080), both NRPE5b and 5c showed tissue are specific expression pattern, with NRPE5c most limited only to silique and flower buds. [S Lahmy et al. *Proc Natl Acad Sci U S A* (2009)] Would the cell line used have a unique NRPE5c specific expression pattern? If NRPE5a~c are all expressed comparatively in the T87 cell line and NRPE5c is selectively assembled into the FLAG-tagged Pol V, what could be the reason? Could it be due to constitutive overexpression of the largest subunit? (for example, NRPE5a may be predominantly used by native Pol V for assembly – therefore the overexpressed FLAG-tagged NRPE1 uses NRPE5c instead?).

14. In table S1, a total of 35 peptides were detected and 6 of them are shown as unique. What does

this mean exactly – the unique six peptide sequences represented 35 times in total, or are there other peptide sequences that are common among other NRPE5s?

15. Fig. S4G shows overall agreement between the model and map 4 but it is difficult to investigate each residue. The density-model validation for a few key side-chains of NRPE5c-specific residues, in comparison to NRPE5a, could be shown to support the identity of the subunit variant (like Fig. S10 for Pol V-KTF1 interface).

16. page9 lines 8-9: the two Pol V-specific subunits “in *Arabidopsis thaliana*” should be noted because Pol V subunit composition and usage varies in other plants– for example, Pol IV and Pol V share the same 5th subunit in maize (JR Haag et al. 2014), an NRPE5 unlike NRPB5 of Pol II.

17. NRPE(E)2, NRPE(B/D)3 and NRPE(A/B/C/D)12 subunits are all shared with Pol IV. Would this mean that KTF1-KOW5 can also associate with Pol IV at the same location? If not, what would prevent Pol IV-KTF1 interaction?

18. RDR2 cannot be considered as a “general transcription factor” or a “transcription elongation factor” for Pol IV. It is a partner in a dual polymerase transcription complex. If anything, RDR2 is a Pol IV backtracking factor, not an elongation factor. Please correct these statements.

19. In the section: The KOW5 domain of KTF1 locates at the RNA exit channel of Pol V, page 11, line 11, Fig 4B-4E: How are the H-bonds (dashed lines) defined in these figures? In the model (Pol V_TEC.pdb), one of the highlighted H-bonds between K639 of KTF1-KOW5 and S855 of NRPE(D)2 has a distance of 4.671 Å, which requires relaxing of distance and angle criteria to be detected as a potential H-bond by the hbonds command in UCSF Chimera. In that relaxed condition, several other potential H-bonds (D858 4.044Å and S841 4.825Å) also show up for K639. It is not clear how the K639-S855 H-bond is selected to be shown in Fig. 4E. Including other H-bonds shown in Fig. 4B-4E, please clarify how they were defined and presented.

20. Page 11 line 15; The two-amino acid (SQ) deletion of SPT5 mentioned here is found only in three Brassicaceae (*A. thaliana*, *C. rubella* and *E. salsugineum*) out of 14 plant species in Fig. S11A. Would the *Arabidopsis* Pol V-KTF1-KOW5 interface be unique in Brassicaceae, and might SPT5 interact with Pol V in other plant species through the SQ residues?

21. It would have been helpful to see if mutations of the KTF1-KOW5 residues involved in Pol V-specific interaction impact the specific interaction with Pol V, to support the Cryo-EM structure.

22. Fig 5 depicts a short stretch of GW/WG tails of NRPE1 and KTF1 anchored to AGO4 close to Pol V. Both NRPE1-CTD and KTF1-CTD have long intrinsic disordered regions with many GW/WG motifs: 17 motifs in 353aa and 44 motifs in 762aa regions respectively, both extending from near the Pol V RNA exit channel toward upstream dsDNA. Considering the potential long stretch of these tails, AGO4 and DRM2 might be recruited to either one or both long flexible tails in a wide range of locations relative to transcribing Pol V. A suggestion is to include this possibility in the model.

23. The Wierzbicki et al papers of 2008 (Cell) and 2009 (Nature Genetics), which were the first to identify Pol V transcripts and show that AGO4-siRNA complexes are recruited to the transcripts, also were the first to show that two of the three proteins that were later found to associate within the so-called DDR complex are required for Pol V transcript production. These papers should be cited when discussing the role of DDR complex proteins in Pol V transcription.

Minor comments:

v The authors claim that Pol V has ‘inferior’ RNA extension ability compared to Pol II in both the abstract and the discussion. Please provide references to support this claim.

v In the abstract, the authors stated that ‘The KOW5 domain of KTF1 binds near the RNA exit channel of Pol V, where it can recruit Argonaute proteins to initiate assembly of DNA methylation machinery’. The latter part of this statement is speculation, and not a primary result, and should be stated as such.

v The manuscript uses a great many abbreviations, and these should be spelled out the first time and abbreviated thereafter. For example, Pol V, should be named Nuclear RNA polymerase V or Nuclear DNA-dependent RNA polymerase V.

v There is no marker/ladder in Fig. S1D. What is the method for determining that the primary RNA products in the autoradiogram are approximately 45 nt in length? In addition, please label the position of RNA primer in the autoradiogram.

Typos

The manuscript has many typographical errors that will need a copy editor to correct, including (but not limited_ to:

- p.4 line 1: DRD6 should be RDR6
- p.4 line 11: "The transcribes genomic regions" should be "The transcribed genomic regions:
- P4. line 20: DSM3 sould be DMS3
- P5. line.12: Pol V-RDR2 should be Pol IV-RDR2
- P5. line.15: delete: "is available"
- P6. line.10: pre-meted should be pre-melted?
- p.7 line 10: RNA-RNA hybrid should be DNA-RNA hybrid
- o There are three instances of "RNA-DNA hybrid" and eight "DNA-RNA hybrid". Please be consistent with either one.
- p.11 line.20: between Po V should be between Pol V
- P13. line.2: DMR2 should be DRM2
- Figure citations should be consistent throughout the manuscript. There are lots of "Figs." In the manuscript.
- Materials and Methods/ Plasmids: pLZ009-GEP-NRPE1 should be pLZ009-GFP-NRPE1?
- p.18, line 9 from bottom: "... the preliminary unreported dataset. subjected to 2D classification (N=30, iterations=25)." Is a broken and incomplete sentence. Please fix this, adding the information on low-pass filtering applied to the auto-picking template (ie. 2D class averages of particles from a preliminary unreported dataset, if understood correctly). Also, the auto-picking template info should be included in the Fig. S2A flowchart for clarification (at "RELION auto-picking Extract 2D classification").

Reviewer #2 (Remarks to the Author):

In this manuscript, Zhang et al. reported the cryo-EM structure of the plant RNA polymerase V as a transcription elongation complex (TEC) associated with elongation factor KTF1 and SPT4 (in visible). The study provided the first Pol V structure. Structural comparison of Pol II and Pol IV revealed unique structural features of Pol V that are distinct from other RNA polymerases and explained some of the Pol V-specific roles, including the RNA-extension and DNA-translocating activities, the lack of interaction with general transcription factors. While I support publication of this study, I strongly suggest that the manuscript should be largely improved and polished before publication.

Major concerns:

1. The manuscript mainly discussed structure features of Pol V, which is fine because of the nature of the study. However, it should also be better to integrate the structural finding in this work and previous genetic and biochemical studies, making the manuscript more informative to broad readers, instead of just structural biologist at transcription field. The authors may cite and discuss previous results, especially those related to the major findings in this work.
2. From the data presented in the manuscript, it's not easy to evaluate the quality of cryo-EM map, especially the critical residues discussed in the main text. These regions should be shown in cartoon/stick that well-coved by corresponding cryo-EM maps, indicating that the structural model was correctly built. The figures could be shown in supplementary figures. Otherwise, discussion at the molecular level may not be supported by the experimental data. It is still fine to describe and discuss these regions by docking structural modules (predicted or from other structures) into the cryo-EM maps, which should be clarified in the manuscript.

Minor concerns:

3. The manuscript seems not to be well-prepared. For example, should it be Pol IV-RDR2 in "we have determined the three-dimensional cryo-EM structures of Pol V-RDR2 complex and reported its unique two-RNA polymerase architecture as well as its unprecedent `backtracking-triggered interpolymerase

RNA channeling' mechanism.". The following sentence, "However, due to the lack of structural information of Pol V is available" seems to be "However, due to the lack of structural information of Pol V". A number of typos could be found throughout the manuscript. The authors should carefully check the manuscript and make correction.

4. Introduction section, "In plants," described DNA methylation pathway by Pol IV, which is not directly related to the major focus of this study. The authors may shorten this part and add more description of Pol V if necessary.

Reviewer #3 (Remarks to the Author):

In this paper, Zhang et al. report the cryo-EM structure of a transcription elongation complex of *Arabidopsis thaliana* RNA polymerase V, which is involved in the RNA-directed DNA methylation pathway in plants. Although Pol V shows a similar structure to Pol II, the DNA-binding cleft of the Pol V TEC is much wider than that of the Pol II TEC. The structure also showed the KOW5 domain of KTF1 bound near the RNA-exit channel, similar to the corresponding domain of the Pol II elongation factor Spt5. The unique structural and sequence features of Pol V may explain the incompatibility of Pol V with the Pol II initiation and elongation factors. Overall, this work is technically sound, and has revealed a novel structure of an important enzyme complex. However, there are several concerns that should be addressed.

Major points:

1. The names of the polymerase subunits are quite confusing. The authors can provide a table to summarize the subunit composition of Pol II and Pol V (and Pol IV) and their correspondence.

2. In the structure reported here, the Pol V TEC adopts an open-clamp conformation. In addition, the KTF1 NGN domain, which should bind to the Pol V main cleft, was not observed. The authors should discuss more about why Pol V adopts such an open-clamp conformation. Is it a general property of Pol V, or is it related to the KTF1 binding? Are there structural features (amino acid residues or segments) that can explain why Pol V prefers the open-clamp conformation, compared to other polymerases? It may also be worthwhile to compare the structure with the Pol II structures with open-clamp conformations, such as TFIIS-complexes.

3. It is written, "Arabidopsis KTF1 contains an NGN domain, three KOW domains (KOW1/4/5)". This reviewer is interested in whether the KOW domains 2, 3 and X are missing in KTF1. According to the AlphaFold prediction, there could be at least one more KOW domain present. For the KOW domains, the authors may refer to the mammalian Pol II-DSIF complex (doi:10.1038/nsmb.3465) or more recent yeast Pol II-elongation factor complexes (doi:10.1126/science.abp9466), as they exhibit more complete KOW domains.

Minor points:

1. Overall, there are many typographical and grammatical errors. Especially, typos concerning the names of Pol IV/Pol V are quite confusing. These should be carefully checked and corrected.

For example:

"In our previous work, we have determined the three-dimensional cryo-EM structures of **Pol V**-RDR2 complex"

"NTP incorporation efficiency and processivity than that of **Po IV**"

"In summary, our structure shows detailed interface between **Po V**"

2. In Fig. S1B, the Pol V preparation contains a lot of impurities. Is it pure enough for the transcription assays or structural analysis? Were the impurities removed after the complex formation and the gel filtration for the structural analysis? Please provide a chart and an SDS-PAGE for the gel filtration step. Also, did the authors observe stoichiometric binding of KTF1?

3. In Fig. S1D, the RNA band positions do not appear to match the sequence diagram on the left side.
4. In the method section, it is written, "The AlphaFold2-predicted structure model of Pol V was used as the start model for model building". What about the KTF1 part? Is it also made by AlphaFold2?
5. Methods and Table S2
300 keV should be 300 kV.

Reviewer #1 (Remarks to the Author):

Using cryo-EM single-particle analysis, this important study by Zhang/Huang/Gu et al. shows the first structure of plant-specific nuclear DNA-dependent RNA Polymerase V (Pol V) bound by Kyrpides-Ouzounis-Woese (KOW) domain-containing transcription factor 1 (KTF1; also known as SUPPRESSOR OF TY INSERTION 5-LIKE, SPT5L), together with a nucleic acid scaffold composed of RNA, template DNA, and non-template DNA to mimic the RNAs of an elongation complex. The overall architecture of Pol V with the DNA/RNA scaffold is very similar to that of Pol II. Unique amino acid compositions in catalytic motifs such as the bridge helix and trigger loop may explain characteristics of Pol V transcription activity reported previously by others. The Pol V-specific N-terminal extension of the 5th subunit (NRPE5) contributes to the unique interaction interface with NRPE1 (the largest subunit), explaining the subunit and polymerase-specificity of the interaction. Also, the identification of the KOW5 domain of KTF1 in the cryo-EM map confirms that its binding location is similar to that of SPT5 when interacting with Pol II. A few Pol V-KTF1 specific differences in the interacting amino acids may account for the Pol V specificity of the interaction. The KTF1-KOW5 domain is located near the RNA exit channel, which would place Glycine-Tryptophan (GW)-rich repeats of KTF1 in the vicinity of Pol V nascent transcripts. These GW repeats of KTF1 and the GW repeats of the Pol V C-terminal domain were shown to be genetically redundant by Lagrange and colleagues in 2016 (see citation information below) and serve as so-called AGO hooks for binding AGO4. Moreover, a new paper by Wang et al. (see citation information below) has shown that ARGONAUTE 4 (AGO4) slices and retains its target RNAs in vivo and in vitro, with prior studies showing that Pol V is the source of the target RNAs in vivo. Collectively, these findings, which are not currently cited, together with the finding that AGO4 can physically interact with DOMAINS REARRANGED METHYLTRANSFERASE 2 (DRM2) (Zhong et al., cited as reference 39 in the current manuscript) support the model the authors present at the end of the paper. The new cryo-EM structure provides the structural context for this model and a basis for future studies exploring how additional proteins associate with the complex to bring about DNA methylation.

Lahmy, S., Pontier, D., Bies-Etheve, N., Laudie, M., Feng, S., Jobet, E., Hale, C.J., Cooke, R., Hakimi, M.A., Angelov, D., et al. (2016). Evidence for ARGONAUTE4-DNA interactions in RNA-directed DNA methylation in plants. *Genes Dev* 30, 2565-2570. 10.1101/gad.289553.116.
Wang, F., Huang, H.Y., Huang, J., Singh, J., and Pikaard, C.S. (2023). Enzymatic reactions of AGO4 in RNA-directed DNA methylation: siRNA duplex loading, passenger strand elimination, target RNA slicing, and sliced target retention. *Genes Dev.* 10.1101/gad.350240.122.

Reply: We thank the referee for the encouraging comments and recommendation. The mentioned references have been cited in the revised manuscript.

Major comments/critiques:

Q1. The term “transcription elongation complex (TEC)” for this study needs to be clarified. The method for “TEC” assembly is derived from other studies with well-characterized RNAPs such as Pol II. But no RNA synthesis and RNA elongation by nucleotide incorporation is demonstrated for the assembled RNA-DNA scaffold, thus it is not clear that the complex is truly an elongation complex in which the synthetic RNA can be extended.

Reply: Thanks for the comment. As pointed out by the referee that the method for assembly of “Pol V TEC” in our study has been used in structural studies of transcription elongation complexes of well-characterized RNAP, including bacterial RNAP, Pol I, Pol II, and Pol III. The pre-melted DNA-RNA scaffold restrains RNAP in a pre-defined translocation state that maximally reduces the conformational heterogeneity and permits structure determination at high resolutions. However, the pre-melted DNA-RNA scaffold is not ideal for testing the activity of elongating RNA as the inability of rewinding the upstream DNA hinders forward translocation of RNAP. Therefore, we employed another DNA-RNA scaffold with a fully complementary non-template and template DNA to show that the purified Pol V sample can extend RNA (Supplementary Fig. 1d).

Q2. Is KTF1 truly an elongation factor? A study in 2009 by He et al, (An effector of RNA-directed

DNA methylation in Arabidopsis is an ARGONAUTE 4- and RNA- binding protein. Cell 137: 498-508) identified a KTF1 mutant they called DMS3. They examined Pol V transcript levels at several loci in *dms3/ktf1* mutants and found that Pol V transcript levels actually increased in the mutant, not decreased as expected if DMS3/KTF1 is important for transcript production. Pol V transcript levels are also increased in *ago4* mutants, probably because AGO4 slicing of the transcripts does not occur. So, KTF1 function may only be for AGO4 recruitment, without affecting Pol V transcription at all. The authors should discuss this and perform an experiment to test whether KTF1 has any effect on Pol V transcription elongation in vitro. KTF1 is SPT5-LIKE, not SPT5 itself, so it may not have the same function as SPT5, as the authors assume.

Reply: Thanks for the comment. We have renamed KTF1 as a AGO4/6-recruiting factor in the revised manuscript. We agree that it is interesting to know whether KTF1 affects transcription elongation (*i.e.* NTP addition, RNAP translocation, pause, backtracking) beyond its recruitment function, the proposed experiments are expected to take at least two months (one month for ordering radiochemical NTPs in China and an additional month for the experiments), but unfortunately we don't have enough time. We are afraid of losing novelty of our story due to the recently published paper of Pol V elongation complex (PMID: 36893216). The suggested experiments will be performed and summarized in a separate manuscript. We have included below words in the introduction section of the revised manuscript.

*"It is intriguing that Pol V transcript levels were barely affected in the *rdm3* mutant²⁸, raising the possibility that KTF1 might not affect the elongation of Pol V but simply functions as a AGO4/6-recruiting factor."*

Q3. A catalytic mutant of NRPE1 should ideally have been used as a control in the transcription assays to prove that the elongation products in Fig. S1D are made by Pol V and not by a contaminating RNA polymerase (e.g., Pools I, II, and III). Including α -amanitin in the transcription assay would at least suppress Pol II's activity, and this is an easy experiment to include.

Reply: Thanks for the suggestion. Our method for Pol V purification includes three steps, an affinity-purification chromatography step, an ion-exchange chromatography step, and a size-exclusion chromatography step, which maximally reduced the chance of contamination of other polymerases. The mass spectrometry result did not detect the largest subunits of other polymerases, suggesting no contamination of Pol I, II, and III. Moreover, the calculated map of Pol V doesn't show any signal for the foot domain of Pol II-NRPB1 subunit and additional subunits of Pol I and Pol III. In summary, we are confident that the extensively purified Pol V doesn't contain any contamination of other three RNA polymerases. It is nice to have an experiment with either α -amanitin or a catalytic-mutant NRPE1 as the control, but we do not have time to perform the experiments due to the reason explained above.

Q4. Page 6 line 12: KTF1/SPT4 was added to "increase stability", but no data showing that KTF1/SPT4 addition indeed improves Pol V stability is presented in the manuscript. A gel filtration profile of the purified Pol V with or without KTF1/SPT4 would be informative.

Reply: We have no evidence showing the KTF1-SPT4 complex stabilizes Pol V TEC, although it was reported SPT5-SPT4 stabilizes Pol II TEC. We have removed such description in the revised manuscript.

Q5. What is the reason for using SPT4-2 (AT5G63670), and not SPT4-1 (AT5G08565)?

Reply: The recombinant SPT4-2 has a much higher expression level than the recombinant SPT4-1 in *E. coli*.

Q6. What is the basis for recombinant KTF1/SPT4 forming a "stable heterodimer"? Are KTF1 and SPT4 stoichiometric in Fig S1?

Reply: The conclusion of the stable dimerization of SPT4 and KTF1 is mainly based on the result that the tag-free SPT4 was stably associated with the his-tagged KTF1 throughout two purification steps. The stoichiometric ratio of KTF1 and SPT4 is expected to ~1:1, but it cannot be precisely determined based on Supplementary Fig. 1. We have performed an additional size-exclusion

chromatography step for the SPT4-KTF1 complex, and the result confirmed stable dimerization of the two proteins. The result has been added into Supplementary Fig. 1f-g.

Q7. The authors state that KTF1/SPT4 forms a stable heterodimer, but this is based only on their co-purification along with other proteins (Fig. S1C), not a demonstration of a direct interaction. The authors should perform reciprocal co-immunoprecipitation or pull-down experiments to confirm the existence of the KTF1/SPT4 heterodimer. Size-exclusion chromatography (SEC) could also be performed to try to isolate and confirm the KTF1-SPT4 heterodimer.

Reply: We have performed an additional size-exclusion chromatography (SEC) and the result confirms stable dimerization of the two proteins (Supplementary Fig. 1f-g).

Q8. The authors refer to “endogenous” Pol V, but this is a bit confusing given that they have isolated epitope-tagged Pol V by virtue of an engineered epitope tag (FLAG-10xHis) on the largest subunit NRPE1. This recombinant subunit is constitutively overexpressed from a strong, non-native 35S promoter, and the transgene is potentially inserted in the genome multiple times. The impact of skewing subunit abundance by constitutive overexpression of only one of 12 subunits on the Pol V assembly pathway is not known. Since the transgenic line was established in the wild-type background, the T87 cell line should also contain Pol V complexes assembled with native NRPE1 subunit, which is the truly endogenous form of Pol V. Perhaps the authors should refer to their complex as “in vivo assembled” or “epitope-tagged” Pol V.

Reply: Thanks for the suggestion. The description of “endogenous Pol V” has been replaced with the “epitope-tagged Pol V”.

Q9. Table S1: As is, the Table S1 title should be “Pol V subunits detected by LC-MS/MS” because it does not describe any of the other proteins that must have been detected in the analysis of the impure sample. There are a great many unidentified proteins in the SDS-PAGE gel shown for the Pol V complex. It would be useful to have the full mass spec report for the proteins present in the fractions provided as a supplemental table.

Reply: The title of the Supplementary Table 2 has been replaced by “Pol V subunits detected by LC-MS/MS of affinity purified epitope-tagged Pol V from *At* T87 cells” in the revised manuscript. The full mass spec report has been submitted as Supplementary Data 1.

Q10. Related to points 9 and 10, does the FLAG tag purified Pol V complex include KTF1 or SPT4? How many replicates did the authors use in the IP-MS experiment, and what is the negative control (e.g. IP of extracts from non-transgenic plants)?

Reply: Similar results were obtained from two replicates (please see below). No KTF1 or SPT4 peptides were detected in our mass spectrometry results. The mass-spectrometry data were collected from the epitope-tagged Pol V sample that was purified through three purification steps from ~5 kg modified T87 cells. The main purpose of the mass-spectrometry experiment is to confirm the presence of Pol V in the purified sample, and therefore no negative control was used.

We have made a mistake when we interpreted the mass mass-spectrometry data in the original manuscript, which has been fixed in the revised manuscript. (see reply to Q12 and Q13).

AGI code	Protein	Epitope-tagged Pol V (tagged at the N-terminus of NRPE1)	Epitope-tagged Pol V (tagged at the C-terminus of NRPE1)
AT2G40030	NRPE1	√	√
AT3G23780	NRPE(D)2	√	√
AT2G15430	NRPE(B/D)3a	√	—
AT2G15400	NRPE(B/D)3b	—	√
AT4G15950	NRPE(D)4	√	—
AT3G57080	NRPE5a	√	√
AT3G54490	NRPE5c	—	√
AT5G51940	NRPE(A/B/C/D)6a	—	—
AT4G14660	NRPE7	√	√
AT1G54250	NRPE(A/B/C/D)8a	—	√
AT3G16980	NRPE(B/D)9a	√	√
AT4G16265	NRPE(B/D)9b	√	√
AT1G11475	NRPE(A/B/C/D)10	√	√
AT3G52090	NRPE(B/D)11	√	√
AT5G41010	NRPE(A/B/C/D)12	—	—

Q11. page.7 lines 15-17: The phrase "including NRPE1 residues R325, P416, D451, A773, K780, Q869, and NRPE(D)2 residues K215, K244, R420, Q725, T740, R806 (Fig. 2A)" is confusing because P416, D451, A773, Q869, Q725, T740 are not positively charged. Also, in the model, D453(E1), rather than D451(E1), appears to make interaction with i-1 U. K921(E2) also seems to interact with i-1 U but this is not included in Fig. 2A? What are the criteria used for claiming specific amino acid interactions?

Reply: We assigned the H-bonds based on the distance (≤ 3.5 Å), geometry, as well as the map quality. We assigned van der Waals interactions mainly based on the distance (≤ 4.5 Å). The ones in the model show reasonable distance and geometry but with poor map were not included. We have rechecked the model, corrected the distance and geometry for those confident H-bonds, and removed the residues with poor map signals in the contact list in the revised manuscript.

The description has been revised as follows,

“The phosphate backbones of the RNA-DNA hybrid make polar and van der Waals interactions with residues in the active-site main cleft, most of which are conserved in Pol II, Pol IV, and Pol V, including NRPE1 residues R325, P415, P416, D451, D453, H456, L772, A773, T777, K780, and NRP(D/E)2 residues K215, R240, F344, R454, S457, N477, N527, K721, Q724, Q725, R806, K921, and H1057 (Fig. 2a).”

Q12. It is intriguing that the purified Pol V from the transgenic T87 Arabidopsis cells has NRPE5c (AT3G54490) exclusively (according to the LC-MS/MS, Table S1), and not NRPE5a (AT3G57080) that have been detected as the predominant Pol V 5th subunit in other studies from multiple labs*. In this work, the cryo-EM single-particle analysis was able to detect subunits that were not found in LC-MS/MS. Because the structural basis of NRPE5-specific Pol V association is a major part of this manuscript, the identity of the 5th subunit in the reconstruction would need to be clarified further. Examples of these studies are Y Li et al. Plant Cell (2018); G Qin et al. Sci Rep (2017); JA Law et al. Curr Biol (2010); TS Ream et al. Mol Cell (2009). This point should be clarified and discussed in the main text, in addition to the legend of Fig. S5.

Reply: Thanks for pointing out the issue. We rechecked the mass spectrometry results carefully and identified that one of the proteins without annotation is NRPE5a, suggesting that both NRPE5a and NRPE5c are present in the epitope-tagged Pol V. However, our current map could not distinguish the two subunits. Due to NRPE5a is more relevant to physiological function, we have replaced NRPE5c with NRPE5a in our revised structure model. The text was also modified as follows,

“LC-MS/MS analysis of the purified complex shows the presence of nine Pol V subunits (Supplementary Table 2), in line with the previous identification of Pol V subunits from Arabidopsis callus except that NRPE5c is also identified in our epitope-tagged Pol V besides NRPE5a⁴². The discrepancy is likely due to different experimental materials used for Pol V preparation.”

Q13. Related to the point above, one wonders if in the cultured cell line T87, how NRPE5a~c expressed? Compared to NRPE5a(AT3G57080), both NRPE5b and 5c showed tissue are specific expression pattern, with NRPE5c most limited only to silique and flower buds. [S Lahmy et al. Proc Natl Acad Sci U S A (2009)] Would the cell line used have a unique NRPE5c specific expression pattern? If NRPE5a~c are all expressed comparatively in the T87 cell line and NRPE5c is selectively assembled into the FLAG-tagged Pol V, what could be the reason? Could it be due to constitutive overexpression of the largest subunit? (for example, NRPE5a may be predominantly used by native Pol V for assembly – therefore the overexpressed FLAG-tagged NRPE1 uses NRPE5c instead?).

Reply: Reanalysis of the IP-MS result shows the presence of both NRPE5a and NRPE5c. Sorry for the mistake. Due to NRPE5a is more relevant to physiological function, we have replaced NRPE5c with NRPE5a in our revised structure model.

Q14. In table S1, a total of 35 peptides were detected and 6 of them are shown as unique. What does this mean exactly – the unique six peptide sequences represented 35 times in total, or are there other peptide sequences that are common among other NRPE5s?

Reply: The peptide sequence that is unique to a certain proteome and can specifically distinguish it from other proteomes is called an unique peptide. We have listed both information for NRPE5a and NRPE5c.

Q15. Fig. S4G shows overall agreement between the model and map 4 but it is difficult to investigate each residue. The density-model validation for a few key side-chains of NRPE5c-specific residues, in comparison to NRPE5a, could be shown to support the identity of the subunit variant (like Fig. S10 for Pol V-KTF1 interface).

Reply: The resolution of map 4 is not good enough to confidently distinguish NRPE5a and NRPE5c. Because reanalysis of the mass spectrometry data reveals the presence of both NRPE5a and NRPE5c, we have replaced the NRPE5a by NRPE5c in the Pol V model as described above in the revised manuscript.

Q16. page9 lines 8-9: the two Pol V-specific subunits “in Arabidopsis thaliana” should be noted because Pol V subunit composition and usage varies in other plants– for example, Pol IV and Pol V share the same 5th subunit in maize (JR Haag et al. 2014), an NRPE5 unlike NRPE5 of Pol II.

Reply: Thanks. It has been corrected in the revised manuscript.

Q17. NRPE(E)2, NRPE(B/D)3 and NRPE(A/B/C/D)12 subunits are all shared with Pol IV. Would this mean that KTF1-KOW5 can also associate with Pol IV at the same location? If not, what would prevent Pol IV-KTF1 interaction?

Reply: We cannot draw a clear conclusion whether KTF1-KOW5 is also able to bind Pol IV at the same location of Pol V. Although we do not observe signals in our cryo-EM map, the other domains (*i.e.* NGN and KOW1) of KTF1 may also interact with the largest subunit of Pol V in some circumstance. Because the interface is unclear of the above domains of KTF1 and NRPE1, we don't know whether the degree of difference on interface residues allows binding of KTF1 with Pol IV.

Q18. RDR2 cannot be considered as a “general transcription factor” or a “transcription elongation factor” for Pol IV. It is a partner in a dual polymerase transcription complex. If anything, RDR2 is a Pol IV backtracking factor, not an elongation factor. Please correct these statements.

Reply: Thanks for the suggestion. RDR2 has been renamed as the backtracking factor in the revised manuscript.

Q19. In the section: The KOW5 domain of KTF1 locates at the RNA exit channel of Pol V, page 11, line 11, Fig 4B-4E: How are the H-bonds (dashed lines) defined in these figures? In the model (Pol V_TEC.pdb), one of the highlighted H-bonds between K639 of KTF1-KOW5 and S855 of

NRPE(D)2 has a distance of 4.671 Å, which requires relaxing of distance and angle criteria to be detected as a potential H-bond by the hbonds command in UCSF Chimera. In that relaxed condition, several other potential H-bonds (D858 4.044Å and S841 4.825Å) also show up for K639. It is not clear how the K639-S855 H-bond is selected to be shown in Fig. 4E. Including other H-bonds shown in Fig. 4B-4E, please clarify how they were defined and presented.

Reply: Due to the moderate resolution of the map, it is not possible to precisely model the side chains and assign the H-bonds. We assigned the H-bonds based on the distance (≤ 3.5 Å), geometry, as well as the map quality. The ones in the model show reasonable distance and geometry but with poor map were not included. We have rechecked the model and corrected the distance and geometry for the confident H-bonds. In the revised model, the H-bond between K639 and S855 has a distance of 2.96 Å; the H-bond between H644 and N853 has a distance of 3.39 Å; the H-bond between R627 and S841 has a distance of 3.38 Å; the H-bond between S642 and Q47 has a distance of 2.91 Å; the H-bond between R615 and Q47 has a distance of 3.30 Å; the H-bond between Q643 and E49 has a distance of 3.23 Å; the H-bond between Y624 and E49 has a distance of 3.21 Å; the H-bond between R617 and N94 has a distance of 2.89 Å; the H-bond between R617 and C93 has a distance of 3.32Å; the H-bond between R615 and D96 has a distance of 2.68 Å; the H-bond between K622 and R170 has a distance of 3.21 Å; the H-bond between K622 and E91 has a distance of 3.09 Å; and the H-bond between Y624 and H51 has a distance of 3.33 Å.

Q20. Page 11 line 15; The two-amino acid (SQ) deletion of SPT5 mentioned here is found only in three Brassicaceae (*A. thaliana*, *C. rubella* and *E. salsugineum*) out of 14 plant species in Fig. S11A. Would the Arabidopsis Pol V-KTF1-KOW5 interface be unique in Brassicaceae, and might SPT5 interact with Pol V in other plant species through the SQ residues?

Reply: Further biochemical or structural studies are needed to determine whether the Arabidopsis Pol V-KTF1-KOW5 interface is unique in Brassicaceae. Although the KOW5 domain of SPT5 might be able to interact with Pol V in other plant species, the other domains of SPT5 might prevent it from interacting with Pol V.

Q21. It would have been helpful to see if mutations of the KTF1-KOW5 residues involved in Pol V-specific interaction impact the specific interaction with Pol V, to support the Cryo-EM structure.

Reply: The cryo-EM map of the interface between KTF1-KOW5 and Pol V is good enough to distinguish KTF1 from SPT5. Therefore, we are confident that KTF1 not SPT5 is present in our structure. Although the requested experiment would provide additional support for the cryo-EM structure, the experiment requires extensive work and time to prepare derivatives of KTF1 and large-quality of Pol V and we are afraid of losing novelty of our story due to the recently published paper of Pol V elongation complex (PMID: 36893216).

Q22. Fig 5 depicts a short stretch of GW/WG tails of NRPE1 and KTF1 anchored to AGO4 close to Pol V. Both NRPE1-CTD and KTF1-CTD have long intrinsic disordered regions with many GW/WG motifs: 17 motifs in 353aa and 44 motifs in 762aa regions respectively, both extending from near the Pol V RNA exit channel toward upstream dsDNA. Considering the potential long stretch of these tails, AGO4 and DRM2 might be recruited to either one or both long flexible tails in a wide range of locations relative to transcribing Pol V. A suggestion is to include this possibility in the model.

Reply: Thanks. The possibility has been included into the model in Fig. 5.

Q23. The Wierzbicki et al papers of 2008 (Cell) and 2009 (Nature Genetics), which were the first to identify Pol V transcripts and show that AGO4-siRNA complexes are recruited to the transcripts, also were the first to show that two of the three proteins that were later found to associate within the so-called DDR complex are required for Pol V transcript production. These papers should be cited when discussing the role of DDR complex proteins in Pol V transcription.

Reply: Thanks. The two literatures have been cited in the revised manuscript.

Minor comments:

Q24. The authors claim that Pol V has ‘inferior’ RNA extension ability compared to Pol II in both the abstract and the discussion. Please provide references to support this claim.

Reply: The citation has been included into the manuscript.

Q25. In the abstract, the authors stated that ‘The KOW5 domain of KTF1 binds near the RNA exit channel of Pol V, where it can recruit Argonaute proteins to initiate assembly of DNA methylation machinery’. The latter part of this statement is speculation, and not a primary result, and should be stated as such.

Reply: The text has been edited based on the suggestion as follows,

“where it provides a scaffold for proposed recruitment of Argonaute proteins to initiate assembly of DNA methylation machinery.”

Q26. The manuscript uses a great many abbreviations, and these should be spelled out the first time and abbreviated thereafter. For example, Pol V, should be named Nuclear RNA polymerase V or Nuclear DNA-dependent RNA polymerase V.

Reply: Thanks for the suggestion. The full names are spelled out for the first time for those abbreviations in the revised manuscript.

Q27. There is no marker/ladder in Fig. S1D. What is the method for determining that the primary RNA products in the autoradiogram are approximately 45 nt in length? In addition, please label the position of RNA primer in the autoradiogram.

Reply: We estimated the RNA products are approximately 40-50 nt in length due to our previous experiences using the same electrophoresis conditions. We have renamed the product as Pol V-extended RNA for accuracy. We are not able to label the position of the RNA primer in the autoradiogram because the RNA primer is not labeled.

Typos

The manuscript has many typographical errors that will need a copy editor to correct, including (but not limited_ to:

- p.4 line 1: DRD6 should be RDR6
- p.4 line 11: “The transcribes genomic regions” should be “The transcribed genomic regions:
- P4. line 20: DSM3 sould be DMS3
- P5. line.12: Pol V-RDR2 should be Pol IV-RDR2
- P5. line.15: delete: “is available”
- P6. line.10: pre-meted should be pre-melted?
- p.7 line 10: RNA-RNA hybrid should be DNA-RNA hybrid
o There are three instances of “RNA-DNA hybrid” and eight “DNA-RNA hybrid”. Please be consistent with either one.
- p.11 line.20: between Po V should be between Pol V
- P13. line.2: DMR2 should be DRM2
- Figure citations should be consistent throughout the manuscript. There are lots of “Figs.” In the manuscript.
- Materials and Methods/ Plasmids: pLZ009-GEP-NRPE1 should be pLZ009-GFP-NRPE1?
- p.18, line 9 from bottom: “... the preliminary unreported dataset. subjected to 2D classification (N=30, iterations=25).” Is a broken and incomplete sentence. Please fix this, adding the information on low-pass filtering applied to the auto-picking template (ie. 2D class averages of particles from a preliminary unreported dataset, if understood correctly). Also, the auto-picking template info should be included in the Fig. S2A flowchart for clarification (at “RELION auto-picking Extract 2D classification”).

Reply: We apologize for the typos/errors in the manuscript. We prepared the manuscript in rush due to potential competition. We have thoroughly checked the manuscript and fixed all the typos pointed by the reviewers as well as others.

Reviewer #2 (Remarks to the Author):

In this manuscript, Zhang et al. reported the cryo-EM structure of the plant RNA polymerase V as a transcription elongation complex (TEC) associated with elongation factor KTF1 and SPT4 (in visible). The study provided the first Pol V structure. Structural comparison of Pol II and Pol IV revealed unique structural features of Pol V that are distinct from other RNA polymerases and explained some of the Pol V-specific roles, including the RNA-extension and DNA-translocating activities, the lack of interaction with general transcription factors. While I support publication of this study, I strongly suggest that the manuscript should be largely improved and polished before publication.

Major concerns:

1. The manuscript mainly discussed structure features of Pol V, which is fine because of the nature of the study. However, it should also be better to integrate the structural finding in this work and previous genetic and biochemical studies, making the manuscript more informative to broad readers, instead of just structural biologist at transcription field. The authors may cite and discuss previous results, especially those related to the major findings in this work.

Reply: Thanks for your suggestion. We have integrated the structural findings and previous genetic and biochemical studies in the following paragraphs,

In the paragraph for describing the interactions between Pol V and DNA/RNA, we added,

“In summary, the Pol V TEC structure show that the active-site cleft of Pol V retains similar interactions with the RNA-DNA hybrid as Pol II, conferring its capacity of RNA extension and DNA translocation (Supplementary Fig. 1d), consistent with the finding the catalytic activity is essential for the function of Pol V in RdDM pathway¹⁶.”

In the paragraph describing the structural motifs in the active-site cleft of Pol V, we stated,

“In short, the sequence and structural comparison of the key structure motifs (bridge helix and trigger loop) in the active site of Pol V, Pol IV, and Pol II provides structural explanation for the differences of transcription activities of the three RNA polymerases.”

In the paragraph describing the wide-open DNA main cleft, we stated,

“It is unclear how the open conformation of Pol V is related to the function of Pol V, but it might partially account for the slower RNA elongation rate of Pol V creating time windows for AGO4/6 recruitment during Pol V transcription elongation.”

Moreover, we use an entire section to describe the structural explanation of the unique function of Pol V by showing that Pol V loses interface for the general transcription factors of Pol IV and Pol II. In the ‘Discussion’ section, we discuss the implication of the KTF1-KOW5 interaction and provide a model for the Pol V-centered DNA methylation pathway.

2. From the data presented in the manuscript, it’s not easy to evaluate the quality of cryo-EM map, especially the critical residues discussed in the main text. These regions should be shown in cartoon/stick that well-coved by corresponding cryo-EM maps, indicating that the structural model was correctly built. The figures could be shown in supplementary figures. Otherwise, discussion at the molecular level may not be supported by the experimental data. It is still fine to describe and discuss these regions by docking structural modules (predicted or from other structures) into the cryo-EM maps, which should be clarified in the manuscript.

Reply: We have prepared the cryo-EM map for the DNA/RNA-interacting residues in Fig. 2g-i. The moderate resolution of map 4 (3.7 Å) does not permit accurate modeling of the side chains of the interface residues of NRPE5 and NRPE1, and therefore the potential interface residues are shown in spheres. Moreover, we removed NRPE1 residue Q869, NRP(D/E)2 residues K244, R245, R247, R420, and Q720, NRPE5 residues T107, K134, and K137 in Fig. 2a due to poor support of the map.

Minor concerns:

3. The manuscript seems not to be well-prepared. For example, should it be Pol IV-RDR2 in “we have determined the three-dimensional cryo-EM structures of Pol V-RDR2 complex and reported its unique two-RNA polymerase architecture as well as its unprecedented ‘backtracking-triggered interpolymerase RNA channeling’ mechanism.”. The following sentence, “However, due to the lack of structural information of Pol V is available” seems to be “However, due to the lack of structural information of Pol V”. A number of typos could be found throughout the manuscript. The authors should carefully check the manuscript and make correction.

Reply: We apologize for the typos/errors in the manuscript. We prepared the manuscript in rush due to potential competition. We have thoroughly checked the manuscript and fixed all the typos pointed by the reviewers as well as others.

4. Introduction section, “In plants,” described DNA methylation pathway by Pol IV, which is not directly related to the major focus of this study. The authors may shorten this part and add more description of Pol V if necessary.

Reply: We have reduced the introduction section of Pol IV and expand the introduction of Pol V subunits and KTF1 function (page 4, paragraph 2; page 5, paragraph 2).

Reviewer #3 (Remarks to the Author):

In this paper, Zhang et al. report the cryo-EM structure of a transcription elongation complex of *Arabidopsis thaliana* RNA polymerase V, which is involved in the RNA-directed DNA methylation pathway in plants. Although Pol V shows a similar structure to Pol II, the DNA-binding cleft of the Pol V TEC is much wider than that of the Pol II TEC. The structure also showed the KOW5 domain of KTF1 bound near the RNA-exit channel, similar to the corresponding domain of the Pol II elongation factor Spt5. The unique structural and sequence features of Pol V may explain the incompatibility of Pol V with the Pol II initiation and elongation factors. Overall, this work is technically sound, and has revealed a novel structure of an important enzyme complex. However, there are several concerns that should be addressed.

Major points:

1. The names of the polymerase subunits are quite confusing. The authors can provide a table to summarize the subunit composition of Pol II and Pol V (and Pol IV) and their correspondence.

Reply: Thanks for your suggestion, we have provided a supplemental table (Supplementary Table 1) for comparison of subunits of Pol II, IV, and V.

2. In the structure reported here, the Pol V TEC adopts an open-clamp conformation. In addition, the KTF1 NGN domain, which should bind to the Pol V main cleft, was not observed. The authors should discuss more about why Pol V adopts such an open-clamp conformation. Is it a general property of Pol V, or is it related to the KTF1 binding? Are there structural features (amino acid residues or segments) that can explain why Pol V prefers the open-clamp conformation, compared to other polymerases? It may also be worthwhile to compare the structure with the Pol II structures with open-clamp conformations, such as TFIIIS-complexes.

Reply: We have added a paragraph in the manuscript to address the question, please see below,

“Our structure reveals that the DNA main cleft of KTF1-bound Pol V TEC adopts a much more opened conformation compared that of Pol II TEC (Fig. 1g, 1h, and Supplementary Fig. 4). The wide-open conformation is not attributed to KTF1 binding, as the recently reported structure of Pol V TEC without KTF1 also shows a wide-open DNA main cleft⁵⁰. Structural comparison between our Pol V TEC and Pol II TEC shows that the clamp domain of Pol V-NRPE1 subunit makes much less interactions with the RNA-DNA hybrid and the downstream dsDNA compared with that of Pol II (Supplementary Fig. 4). The clamp domain of Pol V exhibits disordered lid and rudder loops, and therefore loses interaction with the RNA-DNA hybrid. Moreover, the clamp head domain of Pol V doesn't contain the two DNA grippers of Pol II that interact with downstream dsDNA. We infer that DNA binding is not able to induce clamp closure of Pol V due to the loss of above interactions,

resulting in a wide-open DNA main cleft in the Pol V TEC structure. It is unclear how the open conformation of Pol V is related to the function of Pol V, but it might partially account for the slower RNA elongation rate of Pol V creating time windows for AGO4/6 recruitment during Pol V transcription elongation³⁶.”

3. It is written, “Arabidopsis KTF1 contains an NGN domain, three KOW domains (KOW1/4/5)”. This reviewer is interested in whether the KOW domains 2, 3 and X are missing in KTF1. According to the AlphaFold prediction, there could be at least one more KOW domain present. For the KOW domains, the authors may refer to the mammalian Pol II-DSIF complex (doi:10.1038/nsmb.3465) or more recent yeast Pol II-elongation factor complexes (doi:10.1126/science.abp9466), as they exhibit more complete KOW domains.

Reply: The domain illustration of KTF1 in Fig. 1a was adapted from the previous literature and modified based on AlphaFold2 prediction. There are indeed two tandem KOW domains (residues 435-555) between KOW1 and KOW5 of KTF1, but due to historical reasons, the two respective KOW domains in yeast SPT5 are grouped and named as a single KOW4 domain, the two respective KOW domains in human SPT5 are named KOWx and KOW4 respectively. Previous literatures have grouped the two KOW domains of KTF1 as KOW4, therefore, we followed the nomenclature for consistency. We have slightly adjusted the boundaries of KTF1 domains and corrected the boundaries of KOW4 in the revised manuscript.

Minor points:

1. Overall, there are many typographical and grammatical errors. Especially, typos concerning the names of Pol IV/Pol V are quite confusing. These should be carefully checked and corrected.

For example:

“In our previous work, we have determined the three-dimensional cryo-EM structures of **Pol V**-RDR2 complex”

“NTP incorporation efficiency and processivity than that of **Po IV**”

“In summary, our structure shows detailed interface between **Po V**”

Reply: We apologize for the typos/errors in the manuscript. We prepared the manuscript in rush due to potential competition. We have thoroughly checked the manuscript and fixed all the typos pointed by the reviewers as well as others.

2. In Fig. S1B, the Pol V preparation contains a lot of impurities. Is it pure enough for the transcription assays or structural analysis? Were the impurities removed after the complex formation and the gel filtration for the structural analysis? Please provide a chart and an SDS-PAGE for the gel filtration step. Also, did the authors observe stoichiometric binding of KTF1?

Reply: The chart and SDS-PAGE for the gel filtration step has been provided in Supplementary Fig. 1b-c. The major impurities of the sample are chaperones and heat shock proteins, which cause no interference to the structural study, as the extensive 2D classification steps could discard most of the single particles of the impurities. The mass spectrometry results did not detect any signals for other polymerases besides Pol V, and therefore the transcription activity in Supplementary Fig. 1d reflects the activity of Pol V.

3. In Fig. S1D, the RNA band positions do not appear to match the sequence diagram on the left side.

Reply: Sorry for the misleading. We didn't mean to match the sequence diagram with the RNA band. We have revised the figure to avoid the misleading.

4. In the method section, it is written, “The AlphaFold2-predicted structure model of Pol V was used as the start model for model building”. What about the KTF1 part? Is it also made by AlphaFold2?

Reply: Yes. We have revised the method.

5. Methods and Table S2
300 keV should be 300 kV.

Reply: Revised.

REVIEWERS' COMMENTS

Reviewer #1 (Remarks to the Author):

The authors have addressed most of this reviewer's concerns adequately. The revised atomic model along with reanalysis of MS data clarifies the 5th subunit identity. The inclusion of full MS data as Supplementary Data 1 helps readers better understand the material used for the reconstitution. Although several peptides potentially derived from Pols I, II and III are present in the full MS analysis of Pol V (#688-NRPA1, #955-NRPB2 and #1613-NRPC1, respectively), their low abundance makes it unlikely to contribute to the observed primer elongation activity (Fig. S1d). The Arabidopsis thaliana epitope-tagged Pol V structure has several novel features over the recently published cauliflower Pol V structure, such as inclusion of clamp and stalk domains of Pol V, and KTF1-KOW5. It will also allow comparison among Pol V structures from two related species and facilitate structural understanding of conserved features important for RdDM.

Minor suggestions

Fig. 1a. To be consistent with KTF1 and DNA/RNA, apply gray color to regions of Pol V subunits not modeled in the final structure (each subunit has 10~47% not-modeled residues according to the validation report)

L156. Include reference to Supplementary Data 1, in addition to Supplementary Table 2.

L300. confirmational -> conformational

L753. greay -> gray

L755. nulcleic-acid -> nucleic-acid

L761,774,796. superimposition -> superimposition

L763. comparasion -> comparison

L765. depcits -> depicts

L769. labled -> labeled

L777. thiniana -> thaliana

L780. upstreaan -> upstream

L782. phoaphsate -> phosphate

L791. imcompatible -> incompatible

Reviewer #2 (Remarks to the Author):

The authors have addressed all my concerns in the revised manuscript. I have no further questions and support publication of this work.

Reviewer #3 (Remarks to the Author):

1. The authors performed cryoEM analysis on the mixture sample of Pol Vs containing NRPE5a and NRPE5c, and used the NRPE5a sequence for the model building. This point should be clearly stated in the structure determination section and the methods section.

2. The chart and SDS-PAGE for the gel filtration step has been provided in Supplementary Fig. 1b-c.

Would it be possible to provide the gel for the step: "The mixture was subsequently applied to a Superose 6 Increase 10/300 GL column (Cytiva) and KTF1-SPT4-bound Pol V TEC was eluted"? This reviewer is interested in if SPT4/KTF1 is bound to Pol V in a stoichiometric amount in this gel filtration step.

3. We have thoroughly checked the manuscript and fixed all the typos pointed by the reviewers as well as others.

This reviewer believes that there are still some critical typos.

For example (but perhaps not limited to):

“higher NTP incorporation efficiency and processivity than that of **Po IV**”

“A recent report of cryo-EM structures of Brassica oleracea **Po V**”

Reviewer #1 (Remarks to the Author):

The authors have addressed most of this reviewer's concerns adequately. The revised atomic model along with reanalysis of MS data clarifies the 5th subunit identity. The inclusion of full MS data as Supplementary Data 1 helps readers better understand the material used for the reconstitution. Although several peptides potentially derived from Pols I, II and III are present in the full MS analysis of Pol V (#688-NRPA1, #955-NRPB2 and #1613-NRPC1, respectively), their low abundance makes it unlikely to contribute to the observed primer elongation activity (Fig. S1d).

The Arabidopsis thaliana epitope-tagged Pol V structure has several novel features over the recently published cauliflower Pol V structure, such as inclusion of clamp and stalk domains of Pol V, and KTF1-KOW5. It will also allow comparison among Pol V structures from two related species and facilitate structural understanding of conserved features important for RdDM.

Minor suggestions

Fig. 1a. To be consistent with KTF1 and DNA/RNA, apply gray color to regions of Pol V subunits not modeled in the final structure (each subunit has 10~47% not-modeled residues according to the validation report)

L156. Include reference to Supplementary Data 1, in addition to Supplementary Table 2.

L300. confirmational -> conformational

L753. greay -> gray

L755. nuleic-acid -> nucleic-acid

L761,774,796. superimpostion -> superimposition

L763. comparasion -> comparison

L765. depcits -> depicts

L769. labled -> labeled

L777. thiniana -> thaliana

L780. upstrean -> upstream

L782. phoaphsate -> phosphate

L791. imcompatible -> incompatible

Reply: We apologize for the typos/errors in the manuscript. We have thoroughly checked the manuscript and fixed all the typos pointed by the reviewer. We have added the citation to Supplemental Data 1. We further edited Fig. 1a to color the unmodeled regions in gray. We did not change the color of the small disordered internal loops, otherwise it would interfere with domain definition.

Reviewer #3 (Remarks to the Author):

1. The authors performed cryoEM analysis on the mixture sample of Pol Vs containing NRPE5a and NRPE5c, and used the NRPE5a sequence for the model building. This point should be clearly stated in the structure determination section and the methods section.

Reply: Thanks for the suggestion. We have added some words in the manuscript to address the question, please see below,

"Our current map could not distinguish the two subunits and NRPE5a is modeled into the structure."

2. >The chart and SDS-PAGE for the gel filtration step has been provided in Supplementary Fig. 1b-c. Would it be possible to provide the gel for the step: "The mixture was subsequently applied to a Superose 6 Increase 10/300 GL column (Cytiva) and KTF1-SPT4-bound Pol V TEC was eluted"? This reviewer is interested in if SPT4/KTF1 is bound to Pol V in a stoichiometric amount in this gel filtration step.

Reply: We did not run the SDS-PAGE gel for the eluted fractions of KTF1-SPT4-bound Pol V TEC. The yield of the complex was very poor, and therefore we applied all concentrated sample for grid preparation.

3. This reviewer believes that there are still some critical typos.

For example (but perhaps not limited to):

“higher NTP incorporation efficiency and processivity than that of **Po IV**”

“A recent report of cryo-EM structures of Brassica oleracea **Po V**”

Reply: We apologize for the typos/errors in the manuscript. We have thoroughly checked the manuscript and fixed many typos including the ones pointed by the reviewer.